# Marine protected areas promote stability of reef fish communities under climate warming

Lisandro Benedetti-Cecchi [1] ✉, Amanda E. Bates [2], Giovanni Strona[3], Fabio Bulleri [1], Barbara Horta e Costa [4], Graham J. Edgar [5,6], Bernat Hereu [7], Dan C. Reed [8], Rick D. Stuart-Smith[5,6], Neville S. Barrett [5], David J. Kushner[9], Michael J. Emslie [10], Jose Antonio García-Charton [11], Emanuel J. Gonçalves[12] & Eneko Aspillaga [13]

Protection from direct human impacts can safeguard marine life, yet ocean warming crosses marine protected area boundaries. Here, we test whether protection offers resilience to marine heatwaves from local to network scales. We examine 71,269 timeseries of population abundances for 2269 reef fish species surveyed in 357 protected versus 747 open sites worldwide. We quantify the stability of reef fish abundance from populations to meta-communities, considering responses of species and functional diversity including thermal affinity of different trophic groups. Overall, protection mitigates adverse effects of marine heatwaves on fish abundance, community stability, asynchronous fluctuations and functional richness. We find that local stability is positively related to distance from centers of high human density only in protected areas. We provide evidence that networks of protected areas have persistent reef fish communities in warming oceans by maintaining large populations and promoting stability at different levels of biological organization.

Climate change and direct anthropogenic disturbances are threatening global biodiversity[1,2], often leading to the collapse of ecosystems and the reorganization of ecological communities due to geographic shifts and increasing rates of species extirpations and introductions[3,4]. These processes are increasingly compromising key ecological functions and services such as productivity, nutrient cycling and community resilience to environmental fluctuations[5,6]. When appropriately designed and resourced[7], Marine Protected Areas (MPAs) have proven to be strategic management tools, providing marine life with safe harbors from human disturbances[8,9]. By limiting human extractive uses, direct habitat destruction and a range of local stressors, MPAs can provide multiple ecological and socioeconomic benefits. Decades of research have shown that well-enforced MPAs can increase the diversity, abundance, individual body size and reproductive output of fishes and invertebrates compared to unprotected areas[10–13]. Positive human well-being outcomes may result from increased food security,

[1]Department of Biology, University of Pisa, URL CoNISMa, Via Derna 1, Pisa, Italy. [2]Department of Biology, University of Victoria, Victoria, Canada. [3]European Commission, Joint Research Centre, Ispra, Italy. [4]CCMAR, Centre of Marine Sciences, University of Algarve, Building 7, Faro 8005-139, Portugal. [5]Institute for Marine and Antarctic Studies, University of Tasmania, Hobart, Tasmania, Australia. [6]Reef Life Survey Foundation, Battery Point, Tasmania, Australia. [7]Departament de Biologia Evolutiva, Ecologia i Ciències Ambientals, Facultat de Biologia, Institut de Recerca de la Biodiversitat (IRBIO), Universitat de Barcelona, Barcelona, Spain. [8]Marine Science Institute, University of California Santa Barbara, Santa Barbara 93106 CA, USA. [9]Channel Islands National Park, Ventura, CA, USA. [10]Australian Institute of Marine Science, Townsville, Queensland, Australia. [11]Departamento de Ecología e Hidrología, Universidad de Murcia, Campus Espinardo, Murcia 30100, Spain. [12]MARE – Marine and Environmental Sciences Centre, ISPA – Instituto Universitário, Lisbon, Portugal. [13]Instituto Mediterráneo de Estudios Avanzados (IMEDEA, CSIC-UIB), 07190 Esporles, Spain. ✉e-mail: lbenedetti@biologia.unipi.it

enhanced local fisheries and promotion of cultural, recreational and aesthetic values[14].

While many benefits of reducing local stressors are well documented, whether MPAs can provide ecological resilience and increased adaptive capacity to climate change remains unclear[15–17]. In principle, MPAs can buffer communities from large-scale environmental fluctuations by maintaining high species richness and functional diversity[18–20]. Diverse communities are more likely to compensate for species loss and adapt to environmental change through functional redundancy[21]. Owing to a larger portfolio of ecological responses, effectively protected communities are also expected to display greater temporal stability of aggregated variables such as total species abundance, biomass, and productivity[13,18]. Although improving stability is one of the key goals of MPAs, few studies have provided a direct test of this expectation[13,22–24]. Moreover, how stability varies in relation to key attributes of MPAs such as spatial scale and network size is currently unknown.

Growing interest in understanding how ecosystems respond to increasing environmental fluctuations has led to the development of a theoretical framework to quantify stability and the underlying mechanisms at multiple levels of organization, from individual populations to the metacommunity (i.e., a set of local communities connected by dispersal)[25–29]. This framework adopts an intuitive measure of stability commonly used in the biodiversity-ecosystem functioning literature, which is the inverse of the temporal coefficient of variation – i.e., temporal mean of the variable of interest (e.g., species abundance) divided by its standard deviation[30]. Central to this framework is the mathematical relationship that quantifies the stability of an aggregated variable as the product of the average stability of its constituting elements and the degree of asynchrony in their temporal fluctuations[29]. For example, stability in total community abundance at a particular site (hereafter, alpha stability) results from the product of average temporal stability and asynchrony among the species in the community (Fig. 1a). Lower temporal variation of individual species abundances will result in greater stability in total community abundance, whereas asynchronous fluctuations will promote stability because temporal increases in abundance of some species will compensate temporal declines in other species.

When applied to a metacommunity, the framework allows the partitioning of stability and asynchrony into multiple levels of biological organization (Fig. 1b–f). Metacommunity stability (hereafter, gamma stability) is primarily determined by average alpha stability among local communities and by the degree of asynchrony among them (spatial asynchrony) (Fig. 1b). However, recent work has shown that the degree of stability and asynchrony among species in the metacommunity and among populations within species (metapopulations) are also potentially important mechanisms promoting gamma stability[26,31,32] (Fig. 1, c–f). Furthermore, functional richness – the proportion of the multidimensional trait space occupied by the species in a community[33] – is expected to amplify the stabilizing effects of asynchronous species fluctuations at all levels of organization by broadening the portfolio of possible responses to environmental fluctuations[5,30].

We adopted this framework to compare alpha and gamma stability of reef fish communities and the underlying mechanisms between well-enforced MPAs and areas subjected to some form of extractive use (open areas). We compiled a dataset of 71,269 timeseries of population abundances with a minimum length of 5 years from 2269 reef fish species surveyed at 357 MPA and 747 open area sites across 50 Marine Ecoregions (Fig. 2a). First, we provided a high-resolution analysis using all available sites to examine the effect of protection on alpha stability (Table 1). We expected that MPAs would be more stable than open areas owing to greater stability (lower fluctuations) in the abundance of individual species and greater functional richness. In contrast to these straightforward predictions, anticipating the effect of protection on species asynchrony was more difficult. Suppressing extractive activities within MPAs may reduce asynchrony between targeted and non-targeted species, thereby mitigating the positive effect of asynchrony on stability. However, increased strength in species interactions due to higher species abundances and food-web complexity within MPAs[10,11,34,35] may enhance the contribution of asynchrony to alpha stability compared to open areas.

We also tested the hypothesis that MPAs can buffer communities from marine heatwaves and from other direct human pressures not constrained by MPA boundaries (using the proximity to large cities as a proxy) by relating them to alpha stability and its underlying mechanisms (species stability, asynchrony, and functional richness, Table 1). We adopted a standardized approach to quantify marine heatwaves, defined as sea surface temperatures (SST) anomalies that exceed a seasonally varying climatological threshold (the 90th percentile of SST variation calculated over a 30-yr climatological period), for at least 5 consecutive days[36,37] (see Methods for details). We further quantified the sensitivity of reef fish abundance to marine heatwaves inside and outside MPAs using the Species Temperature Index (STI) – a well-known measure of the realized temperature niche of a species[18,38]. Marine heatwaves affect marine biodiversity globally[20,39], thus providing an appropriate synthetic climate variable to evaluate the potential buffering effect of protection on reef fish abundance and stability. Similarly, distance from large human settlements is a suitable predictor of the abundance, richness, and vulnerability of reef fish communities and is thus a suitable surrogate measure of direct human pressure on these communities[40,41].

Finally, we compared stability and asynchrony between MPAs and open areas at the metacommunity level. We considered the sites in an ecoregion as part of a metacommunity and the MPA sites as a spatial network of connected sites based on proximity[42,43]. We expected two opposite mechanisms to affect gamma stability in metacommunities. On the one hand, we hypothesized that environmental heterogeneity would magnify large-scale population and species spatial and temporal fluctuations, weakening their contribution to gamma stability. Thus, by reducing direct human disturbances, MPAs should mitigate population and species fluctuations, increasing gamma stability. On the other hand, environmental variability should increase the contribution of asynchrony to gamma stability at all levels of organization (spatial, species, metapopulations, Fig. 1, b–e). Therefore, we hypothesized that both asynchrony and gamma stability should increase in large MPA networks across a wide range of habitats and environmental conditions[44,45] (Table 1).

Overall, our results indicate that well-enforced MPAs can promote stability of reef fish abundance at the communities and metacommunity levels, mitigating the adverse effects of MHWs in addition to direct human disturbance.

## Results
### Alpha stability
We first examined the relationships between stability components and their predictors using Linear Mixed Effect Models with a random intercept for study ID and including the total area sampled at each site as an offset to control for sampling effort (full results are reported in Supplementary Tables 1–3). Note that by including an offset in the mixed-effect models, we scaled each response variable (alpha and species stability, asynchrony and functional richness) to the total area sampled at each site (see *Methods*, '*Controlling for sampling effort*' for a justification of this approach). Alpha stability was positively related to species asynchrony (measured using the Gross index[46]) and species stability in MPAs and open areas (Fig. 2b, c). In contrast, alpha stability declined with increasing mean intensity of marine heatwaves in open areas, but not in MPAs (Fig. 2d). Similar patterns were observed for species stability (Fig. 2f), species asynchrony (Fig. 2h) and functional

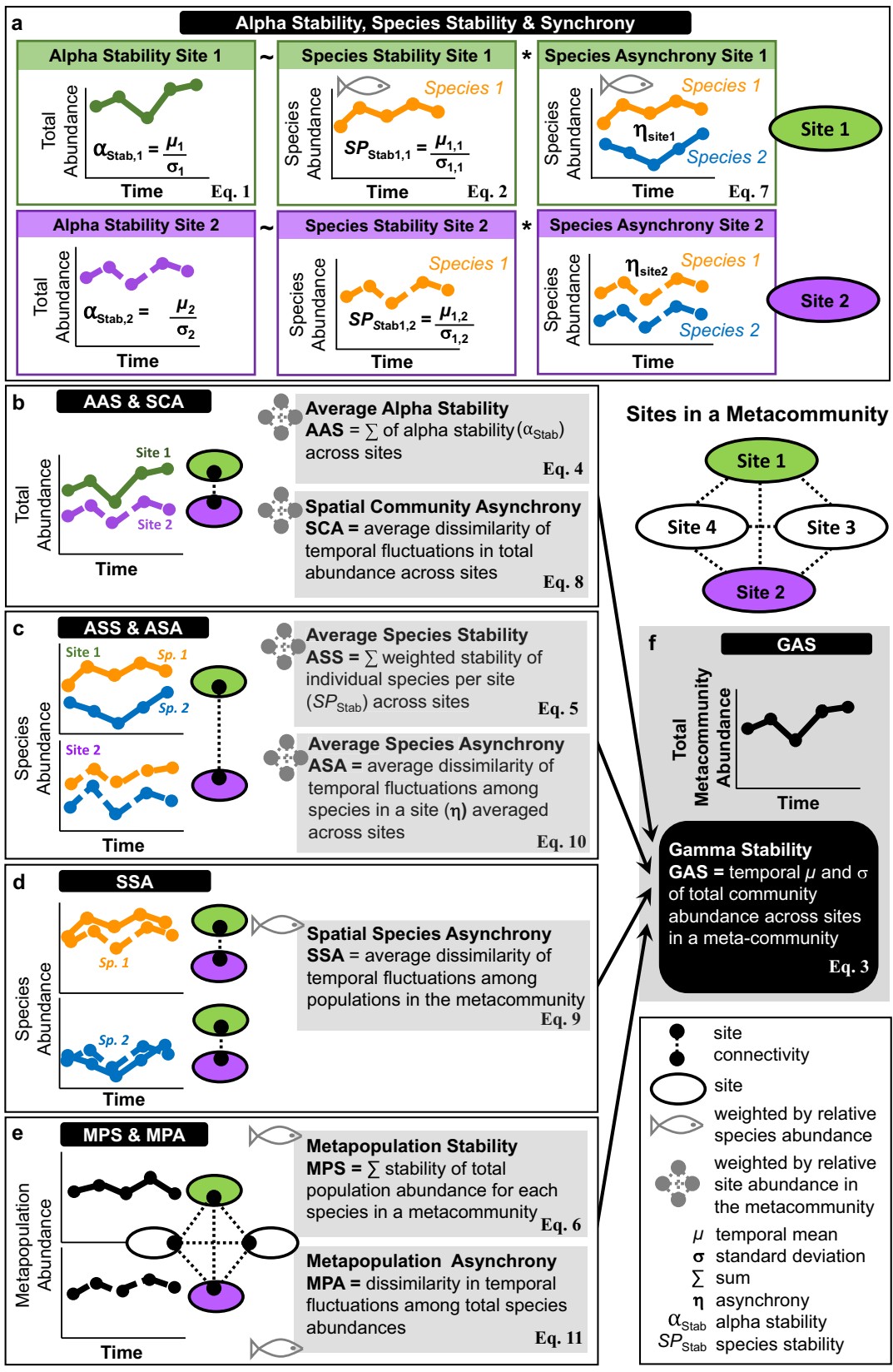

richness (Fig. 2j), all decreasing significantly with marine heatwaves in open areas, but not in MPAs. Alpha stability increased significantly with remoteness in MPAs, but not in open areas (Fig. 2e), whereas species stability was positively associated with remoteness both in MPAs and in open areas (Fig. 2g). Species asynchrony and functional richness were unrelated to remoteness (Fig. 2i, k), although functional richness was higher in MPAs than in open areas at all values of remoteness (Fig. 2k). Functional richness was positively associated with species stability in MPAs and open areas and negatively associated with alpha stability and species asynchrony in open areas (Supplementary Fig. 1, Supplementary Table 1). Alternative mixed-effect models that included interaction terms between protection levels (MPA vs. open areas) and

**Fig. 1 | The stability framework.** Panels illustrate the different components of stability and asynchrony obtained from reef fish abundance data at the community (**a**) and metacommunity (**b**–**f**) levels of organization. Two sites, each including one population of two species, are used throughout to illustrate the derivation of stability and asynchrony measures from timeseries of fish abundance. Stability is indicated as the ratio between the temporal mean and standard deviation of fish abundance ($\mu/\sigma$), whereas $\eta$ indicates asynchrony. **a** Alpha stability, species stability and species asynchrony; $\mu_{i,j}$ and $\sigma_{i,j}$ are the temporal mean and standard deviation of species $j$ at site $i$, respectively. **b** Average alpha stability (AAS) and spatial community asynchrony (SCA) calculated from total fish abundance between two sites. **c** Average species stability (ASS) and average species asynchrony (ASA) calculated from the two populations of each species and then averaged between species.

**d** Spatial species asynchrony (SSA) quantified as the average dissimilarity of temporal fluctuations between populations. **e** Metapopulation stability (MPS) and metapopulation asynchrony (MPAS), calculated from total population abundance and averaged across species. **f** Gamma stability (GAS) obtained by dividing the temporal mean of total metacommunity abundance by its standard deviation. Arrows pointing to this panel indicate the positive contribution of stability and asynchrony at lower organizational levels to gamma stability. Pink and green ovals indicate whether timeseries were aggregated among species within sites (**b**, **c**), among populations in the metacommunity (**d**) or among metapopulations (**e**) to derive stability and asynchrony measures. Panels indicate the equations (Eq) used to calculate the various stability and asynchrony measures, which are described in full in *Methods*.

predictors produced similar results (Supplementary Fig. 2, Supplementary Tables 2, 3).

Results were robust to detrending of timeseries[47] and specific choices of asynchrony measures (Supplementary Fig. 3), as well as to quantification of marine heatwaves (mean vs. cumulative intensity, Supplementary Fig. 4). Alternative analyses based on log-response ratios, where sampling effort was controlled by dividing response variables directly by the total area sampled at each site, rather than through an offset, provided similar results to those of the main analysis (Supplementary Fig. 5). Consistent results were also obtained by excluding monitoring programs that targeted a limited set of species (50 or less), suggesting differences in taxonomic scope among programs did not affect the results (Supplementary Fig. 6). Furthermore, sample coverage, a measure of sampling completeness[48], indicated that fish communities were sampled with comparable accuracy in MPAs and open areas (Supplementary Fig. 7). Only transects in the size category of 180 m² indicated larger completeness in open areas than in MPAs. These transects represented a small fraction (2.2%) of the total samples and removing them from the analysis did not change the results.

## Causal pathways

We used piecewise Structural Equation Modeling (SEM)[49] to explore the causal pathways illustrated in Fig. 1, along with the hypothesized influences of marine heatwaves (quantified through mean intensity) and remoteness. SEMs conducted in MPAs and open areas differed markedly in terms of magnitude, direction, and sign of significant links, with a prevalence of destabilizing effects in the absence of protection (Fig. 3a, b). The most striking difference involved the links connecting marine heatwaves to alpha and species stability, species asynchrony and functional richness. While marine heatwaves had no significant *direct* or *indirect* effects on stability and asynchrony in MPAs (Fig. 3a, c), they destabilized reef fishes in open areas through significant *negative direct* and *indirect* effects on alpha stability and *negative direct* effects on species stability and asynchrony (Fig. 3b, c). Marine heatwaves also had *direct negative* effects on functional richness in open areas, which generated weak *positive and negative indirect* effects on species asynchrony and species stability, respectively (Fig. 3b, c). Remoteness translated into *positive direct* and *indirect* effects on alpha stability in MPAs and open areas, respectively, the latter through species stability (Fig. 3). Overall, all the significant paths pointed to positive effects in MPAs, whereas only 4 of the 10 significant links were positive in open areas (Fig. 3c). SEM results were robust to detrending of timeseries[47] and specific choices of asynchrony measures (Supplementary Fig. 8).

## Thermal sensitivity trends

We performed two additional analyses to explore the mechanisms behind the different impact of marine heatwaves on stability in MPAs and open areas observed in the SEM results. First, we examined how the two components of stability, the temporal mean and standard deviation of total fish abundance, varied in relation to marine

heatwaves. Greater stability may result from larger mean abundance (the numerator of stability), lower standard deviation (the denominator), or a combination of both[50]. We found that changes in stability were driven mainly by variation in the mean rather than in standard deviation, with mean fish abundance increasing with marine heatwaves in MPAs and decreasing in open areas. The standard deviation did not change in MPAs, while it declined with intensifying marine heatwaves in open areas (Supplementary Fig. 9).

Second, we examined whether MPAs could support higher fish abundances and promote stability by allowing thermally resistant species to attain large population sizes under intensifying warming conditions. To test this hypothesis, we defined a thermal threshold based on the maximum intensity of marine heatwaves observed over the sampling period of fish abundance at each site. Then we divided fish species into two groups depending on their Species Temperature Index (STI), a well-known measure of the realized temperature niche of a species[18,38] (see Methods): those with a STI equal to or above threshold (thermally resistant species); and those with a STI below threshold (thermally sensitive species). Our definition of thermal threshold based on the maximum intensity of marine heatwaves provided a more stringent definition of thermally resistant species compared to a threshold based on mean intensity (which we used, instead, as a covariate as in the previous mixed-effect models and SEM analyses). We summed species abundances separately for species with STIs above or below threshold at each site and used these aggregated values to compare thermal sensitivity trends between MPAs and open areas against the mean intensity of marine heatwaves. Previous studies have shown that different trophic groups can respond differently to warming. For example, grazers can benefit from elevated temperatures owing to increased metabolism and faster feeding and digestion rates[18,51]. Thus, we examined thermal sensitivity trends separately for four trophic categories: carnivores, grazers, microphages and planktivores (Fig. 4, Supplementary Table 4). We used Generalized Additive Mixed Models (GAMMs, which included a random effect for study ID) in these analyses to account for the non-linear relationships between the abundance of trophic categories and marine heatwaves. When considering species with STIs above the threshold, all trophic groups showed a peak in abundance at intermediate to high intensities of MHWs in MPAs, but not in open areas (Fig. 4, Supplementary Table 4). A similar pattern was observed for thermally sensitive species (STIs below threshold), although with some exceptions. Abundances of thermally sensitive carnivores and planktivores were only weakly related to marine heatwaves in MPAs, whereas grazers showed a consistent trend of increasing abundance with marine heatwaves both in MPAs and in open areas. The abundance of all other trophic groups generally declined at maximum intensity of marine heatwaves (Fig. 4, Supplementary Table 4).

## Metacommunity networks and connectivity

Most ecoregions included multiple open area and MPA sites, the latter distributed over a single large MPA (e.g., Great Barrier Reef Park) or multiple smaller MPAs (Supplementary Table 5). We considered the

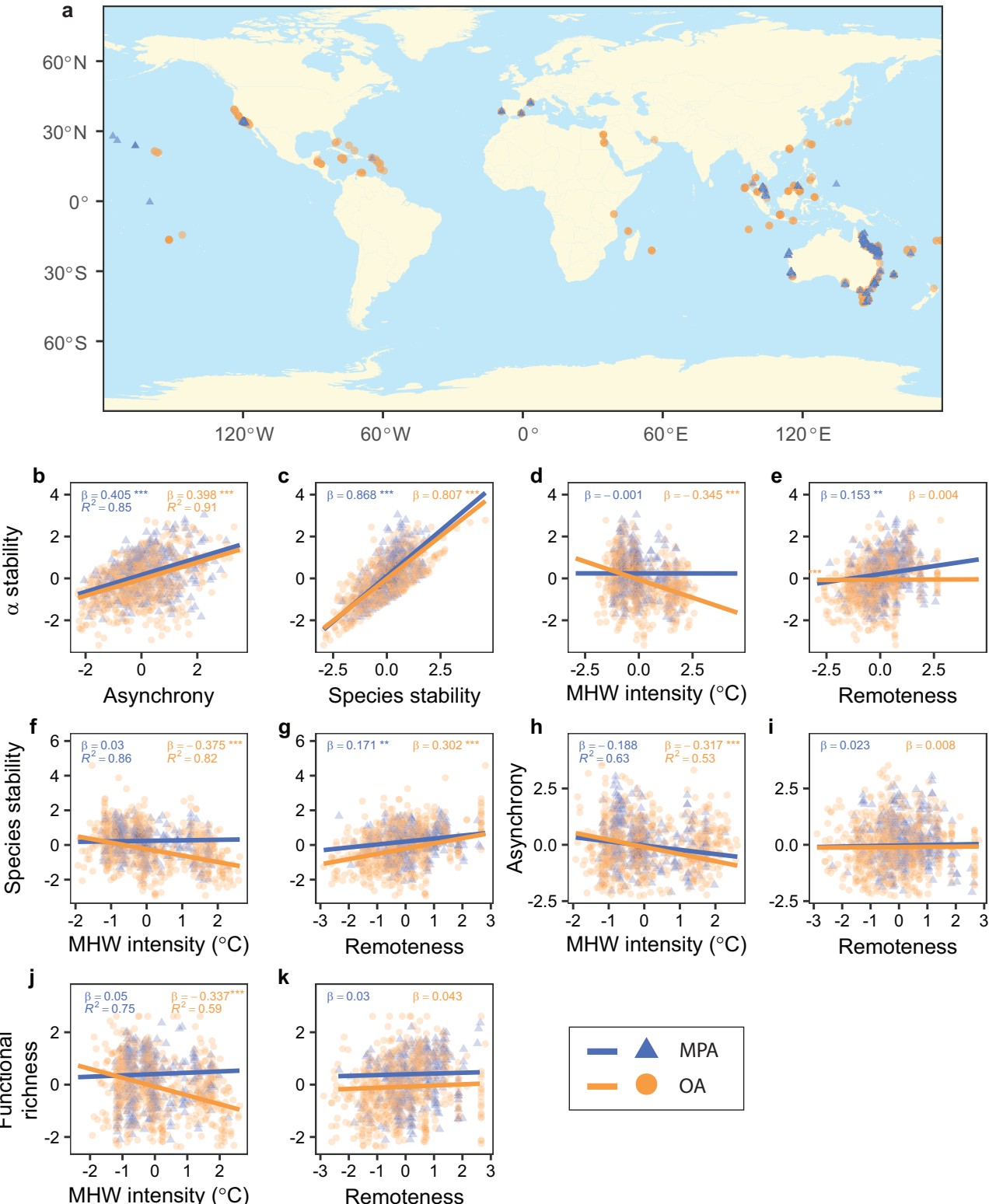

**Fig. 2 | Timeseries of reef fish abundance and derived measures. a** Study sites. **b**–**k** Relationships between alpha stability, species stability, species asynchrony, functional richness, and their hypothesized drivers. Data are shown as z-scores for marine protected areas (MPA) and open areas (OA). Panels include the regression parameters estimated from Linear Mixed Effect Models, their significance (2-tailed $t$-tests; ***$p < 0.001$; **$p < 0.01$) and the conditional coefficients of determination ($R^2$, indicated only in the first panel for each response variable). Full statistical results are reported in Supplementary Tables 1–3.

MPA sites in an ecoregion as part of a metacommunity network, regardless of whether they occurred in one or in several MPAs. Distances among sites ranged from <1 km to about 1000 km and although the largest distance is beyond the direct dispersal range of most, if not all, reef fish species, distant sites may become connected over multiple generations through stepping-stone effects[52].

Although connectivity is rarely assessed in studies of gamma stability[27,28], understanding whether sites are linked by the movement

**Table 1 | Main hypotheses relating effects of Marine Protected Areas (MPA) on site (alpha) stability, species stability and asynchrony, compared to open areas (OA)**

| Hypothesized relationships | Predicted effect | Mechanisms | References |
|---|---|---|---|
| *Within sites (communities)* | | | |
| MPA → alpha stability | MPA are more stable than OA. | Greater stability (lower fluctuations) in abundance of individual species and greater functional richness increases stability in MPA compared to OA. | 13,15–18,22 |
| MPA → species stability → alpha stability | Stronger positive relationship between alpha stability and species stability in MPA than OA. | MPA increase alpha stability by maintaining more stable populations compared to OA. | 13,22–24 |
| §MPA → asynchrony → alpha stability | The contribution of asynchrony to stability is stronger (weaker) in MPA than OA. | MPA increase (decrease) asynchrony if relieve from fishing and direct human disturbances induce divergent (coherent) temporal fluctuations in fishes. | 10,11,34,35 |
| MPA → MHW → species and alpha stability | Alpha and species stability decline more abruptly with intensifying MHW in OA than MPAs. | MPA buffer reef fishes from MHW by maintaining greater population abundances, functional richness and asynchronous fluctuations, all of which contribute to increase stability. | 18 |
| MPA → Remoteness → species and alpha stability | Stronger positive relationship between alpha and species stability with remoteness in MPA than OA. | Remote MPA are relieved from both fishing and direct human impacts, whereas fishing can still impact remote sites in OA, decreasing alpha and species stability. | 40 |
| MPA → temperature niche | Stronger positive relationship between the abundance of thermally resistant species with intensifying MHW in MPA than OA. | MPA support higher fish abundances and promote stability by allowing thermally resistant species to attain large population sizes under intensifying warming conditions. | 18,61,62 |
| *Among sites (metacommunities)* | | | |
| MPA → gamma stability and underlying mechanisms | Greater asynchrony and stability in MPA than OA at the metacommunity scale. | Local effects of MPA on species stability and asynchrony scale-up at the metacommunity level. | 44,45,65 |
| Size of MPA network → gamma stability and underlying mechanisms | Differences between MPA and OA increase with the size of MPA networks. | Gamma stability and the underlying mechanisms operate more strongly in large MPA networks embracing a wide range of habitats and environmental conditions. | 44,45,65 |

MPAs are hypothesized to mediate effects of Marine Heatwaves (MHW) and remoteness (distance from large cities) at local scales. At the metacommunity scale, gamma stability is related to multiple underlying mechanisms, including population and species stability and population, species and spatial asynchrony (Fig. 1).
§There is no clear a priori expectation about the direction of the effect of MPAs on asynchrony and both positive and negative effects are considered here.

of individuals (larvae, juveniles and adults) is important for delineating a metacommunity. Given the limited knowledge on dispersal of reef fishes, we used graph theory to characterize the spatial structure and connectivity of metacommunities[53]. Following previous work, we derived a minimum spanning tree graph using all sites in each ecoregion and quantified centrality metrics on these graphs. A minimum spanning tree graph is a network that uses the minimum number of shortest links to ensure that all nodes (sites) are connected without closed paths among nodes[54]. We derived two networks for each metacommunity, one based on biological distance (community compositional dissimilarity, quantified by the Jaccard index, Supplementary Fig. 10) and another based on geographic distance (least-cost path distance among sites by the sea, Supplementary Fig. 11). We computed two centrality metrics to characterize the topological features of these networks and to extract information on connectivity: degree centrality and closeness centrality[53,55]. In weighted networks, degree centrality is computed, for each node, as the sum of the weights of links connecting the target node to its neighbors, hence providing a measure of local connectivity. Closeness centrality is computed for each node as the reciprocal of the average weighted distance (i.e. shorted cumulative weighted path across network links) from the target node to all other nodes in the network. This provides a measure of global connectivity. Using these metrics, we tested the prediction that, in a dispersal-limited metacommunity, physically isolated sites with low closeness centrality should also be more biologically distinct (high community dissimilarity and low degree centrality) than more central sites in a minimum spanning tree graph. This should result in significant positive relationships between closeness centrality measured on a geographically-derived graph and degree centrality measured on a biologically-derived graph. In contrast, in a well-mixed metacommunity, geographic distance among sites should have no bearing on biological distance. We found no significant relationships between centrality measures in any of the metacommunities analyzed,

suggesting that geographic isolation did not preclude biological connectivity (Supplementary Table 6). Furthermore, average compositional dissimilarity, computed across all sites within each metacommunity using the Jaccard index, ranged between 0.21-0.74 and 0.26-0.64 in MPAs and open areas, respectively, suggesting that fish dispersal neither completely differentiated nor completely homogenized metacommunities (Supplementary Table 5).

### Metacommunity stability and asynchrony

We compare differences in stability and asynchrony measures at the metacommunity level (Fig. 1b–f) between MPA and open area sites in relation to three key attributes of MPA networks: spatial scale, the number of MPAs and the total number of sampled sites. The spatial scale of each MPA network was defined by the maximum distance between MPA sites in an ecoregion and was used to match MPA an open area sites for comparison (*Methods: Metacommunity-level analysis*). Since calculating asynchrony at the metacommunity level required matching timeseries (i.e., data sampled in the same years) among sites, we limited this analysis to a subset of ecoregions that allowed comparisons between at least two MPA and two open area sites, while ensuring a minimum length of timeseries of 5 years.

We calculated the following abundance-weighted stability and asynchrony measures separately for MPA and open area sites within metacommunities (see Fig. 1, b–f and Methods for details): gamma stability, as the inverse of the coefficient of variation of total metacommunity abundance ($TCV^{-1}$); average alpha stability, as the $TCV^{-1}$ of total site abundance (the alpha stability measure used in the previous site-scale analysis) averaged across sites; average species stability, as the average of species $TCV^{-1}$s in a site further averaged across sites; metapopulation stability, as the $TCV^{-1}$ of summed population abundances averaged over species. Similarly, for asynchrony we calculated: spatial community asynchrony, as the average asynchrony in total abundance among sites; spatial species asynchrony, as the average

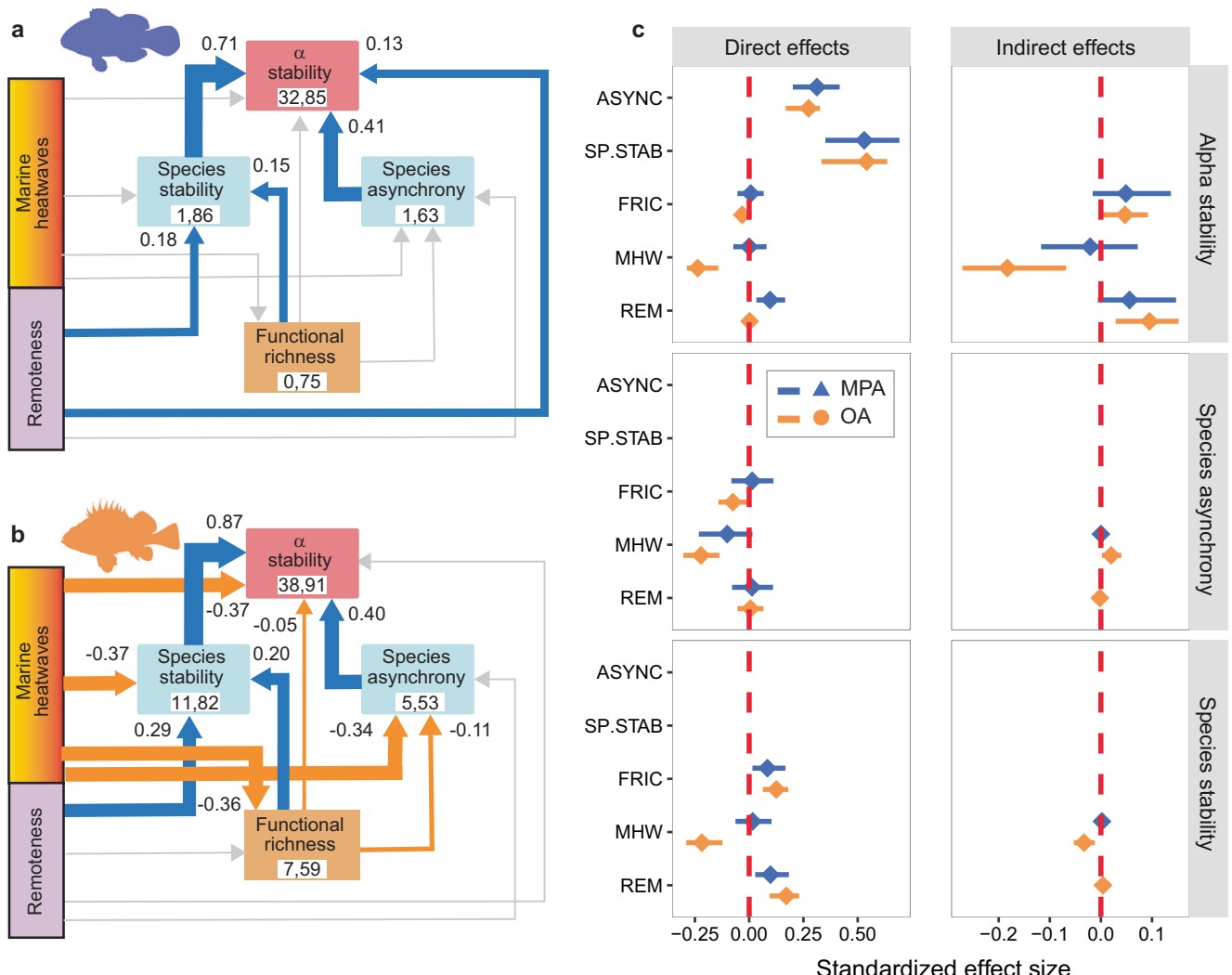

**Fig. 3 | Piecewise Structural Equation Models (SEM) of reef fish alpha stability.** Path diagrams are illustrated for **a** marine protected areas (MPA) and **b** open areas (OA). Positive (negative) links are shown in blue (orange), with path size proportional to the standardized regression coefficient. Not significant paths ($p > 0.05$) are shown in light grey. Numbers within boxes indicate the variance explained by fixed (marginal, left) and total – i.e. fixed and random together – (conditional, right) effects in the model. **c** Standardized direct and indirect effect sizes (means and 95% Confidence Intervals derived from $n = 10,000$ bootstrap replicates of the SEM

model) of factors influencing alpha and species stability and species asynchrony in MPA (blue) and OA (orange). Effect sizes whose confidence intervals do not overlap with zero (dashed red line) are considered significant. Positive (negative) effect sizes indicate larger (lower) stability or asynchrony in MPAs than open areas. ASYNC: species asynchrony; SP.STAB: species stability; FRIC: functional richness; MHW: marine heatwaves intensity; REM: remoteness. The not significant ($p > 0.5$) link from remoteness to functional richness was removed from the original MPA path diagram to improve model fit (Fisher's $C$ statistic: $p > 0.05$ for both models).

asynchrony among populations; average species asynchrony, averaging asynchrony among species in a site (the asynchrony measure used in the previous site-scale analysis) and then over sites; metapopulation asynchrony, as the asynchrony among summed population abundances.

This analysis resulted in a single value of the gamma stability metric (and any other metacommunity measure) for each of the MPA and open area conditions in an ecoregion, precluding the direct estimation of variances and hindering the statistical comparison between these conditions. To overcome this problem, we used a jackknife (leave-one-out) procedure that allowed us to obtain robust estimates of variances and to derive the *Hedge's g* effect size of the difference between MPAs and open areas for each metric, which we analyzed in a Bayesian meta-analytical framework (see Methods).

There was no clear trend of variation in stability and asynchrony measures with spatial scale across the 12 metacommunities examined (Fig. 5, Supplementary Fig. 12). A positive effect size indicating larger gamma stability in MPAs than in open areas was observed in 6

metacommunities, whilst effect sizes either did not deviate significantly from zero or were negative, the latter indicating higher gamma stability in open areas than in MPAs, in the other 6 metacommunities (Fig. 5b). Higher stability in MPAs than in open areas was more evident for the other components of stability, with 8, 9 and 10 of the 12 metacommunities having higher alpha, species and metapopulation stability in MPAs than in open areas, respectively (Fig. 5c–e). Asynchrony measures did not show any consistent difference between MPAs and open areas, with the possible exception of spatial asynchrony, which was greater in open areas than MPAs (negative effect size) in 6 metacommunities, whilst MPAs had large spatial asynchrony in 4 metacommunities (Supplementary Fig. 12). Effect sizes did not vary significantly with the number of MPAs in each network nor with the total number of sites examined (Supplementary Table 7).

To assess the robustness of our results to the choice of the spatial scale over which comparisons were conducted, we repeated the analysis by matching MPA and open area sites within a spatial scale of 50–100 km, instead of using the maximum distance between MPA sites

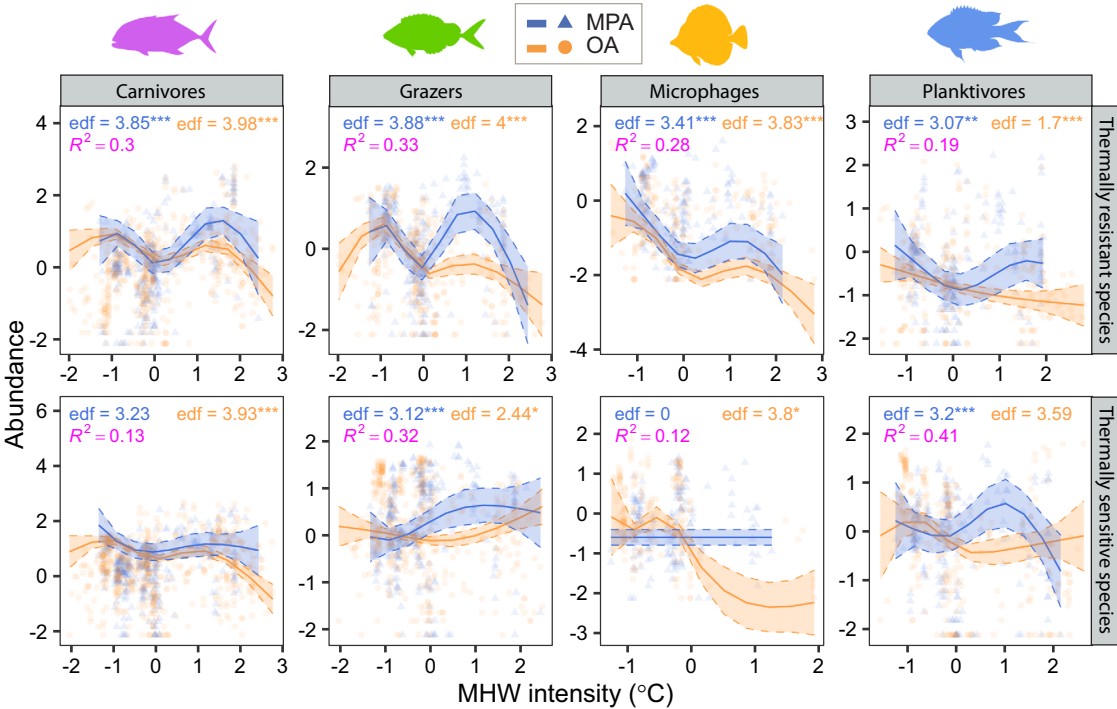

**Fig. 4 | Thermal sensitivity trends of reef fish.** Panels show the trajectories obtained by fitting Generalized Additive Mixed Models (GAMMs) to the abundance (log-transformed and standardized) of four fish trophic categories with thermal affinities below (thermally sensitive species) or equal-above (thermally resistant species) the thermal threshold, against mean intensity of marine heatwaves (MHWs). Thermal thresholds are based on the maximum MHW intensity recorded at a site during the sampling period. Trends are plotted separately for marine protected areas (MPA, blue lines, and symbols) and open areas (OA, orange lines, and symbols); filled areas indicate standard errors; data are shown as z-scores. Panels include GAMM effective degrees of freedom, their significance (***$p < 0.001$; **$p < 0.01$; *$p < 0.05$) and the coefficients of determination ($R^2$). Full statistical results are reported in Supplementary Table 4.

(*Methods: Metacommunity-level analysis*). This range of distances was intermediate between the maximum distances separating MPAs in metacommunities (Supplementary Table 5), with 100 km representing a potential upper limit of direct fish dispersal[40,56]. Results for the six metacommunities that encompassed the 100 km spatial scale were very similar to those observed in the analysis using maximum distances, suggesting that results were not affected by the particular spatial scale at which metacommunity stability and asynchrony were compared (compare Fig. 5b–e and Supplementary Fig. 12 with Supplementary Fig. 13). Null model analysis indicated that asynchrony was lower than expected by chance for most metacommunities, and that differences between MPAs and open areas were highly context-dependent (Supplementary Fig. 14).

## Discussion

To our knowledge, our study provides the first global analysis of whether, to what extent and through which mechanisms MPAs affect reef fish stability in the face of global warming. Our results support the hypothesis that MPAs promote reef fish stability at the community and metacommunity levels. A fundamental question in marine conservation is whether MPAs can mitigate the effects of large-scale climate change and anthropogenic impacts on natural communities. The rationale is that MPAs can promote resilience and adaptation to climate change by sustaining large populations and diverse communities[15,18]. Indeed, our comprehensive analysis of reef fish timeseries suggests that well-enforced MPAs can buffer the impact of marine heatwaves on species and community stability by supporting larger populations, preserving functional richness and maintaining stronger asynchronous fluctuations compared to open areas.

Marine heatwaves are a major threat to the structure and functioning of marine ecosystems and have been associated with extensive and recurrent mass mortality events of marine life and loss of ecosystem services[39,57]. Projections suggest that marine heatwaves will become more pervasive leading to abrupt changes in ocean climate in the next decades[58,59] and that current warming rates will soon exceed the thermal safety margin of many species[16]. A continent-wide evaluation of decadal trends in abundance of reef fishes, corals, invertebrates and algae around the coasts of Australia, showed significant population declines following marine heatwaves, especially near warm range edges and for large-size organisms[60]. Similarly, mass mortality events of reef-dwelling species are increasingly documented in the Mediterranean Sea, which is warming at an alarming rate of three times that of the global ocean[57,61]. Yet, we found a positive relationship between fish abundance and marine heatwaves for most trophic categories in MPAs. Surprisingly, fishes that experienced warming events beyond their upper realized thermal limit (STI below threshold) increased in abundance with maximum intensity of marine heatwaves when protected from direct human disturbance (Fig. 4, lower panels). In contrast, the abundance of most trophic categories declined with intensifying marine heatwaves in open areas, regardless of their thermal sensitivity.

Grazers were the only trophic category that showed a consistent trend of increasing abundance with marine heatwaves in open areas. This outcome was in agreement with previous studies documenting positive responses of herbivorous fishes to warming, in terms of increased population abundance, species richness and tropicalization – i.e. the range expansion of tropical species into temperate regions[18,62,63]. However, only grazers with a STI below the threshold increased in abundance with marine heatwaves in open areas, whereas those with a STI above the threshold declined. Since marine heatwaves are generally more intense away from the tropics[37], a trend of declining abundance for trophic groups with STIs above the threshold could be a consequence of the poleward decline in grazers and tropicalization impacts[63], with warm-water

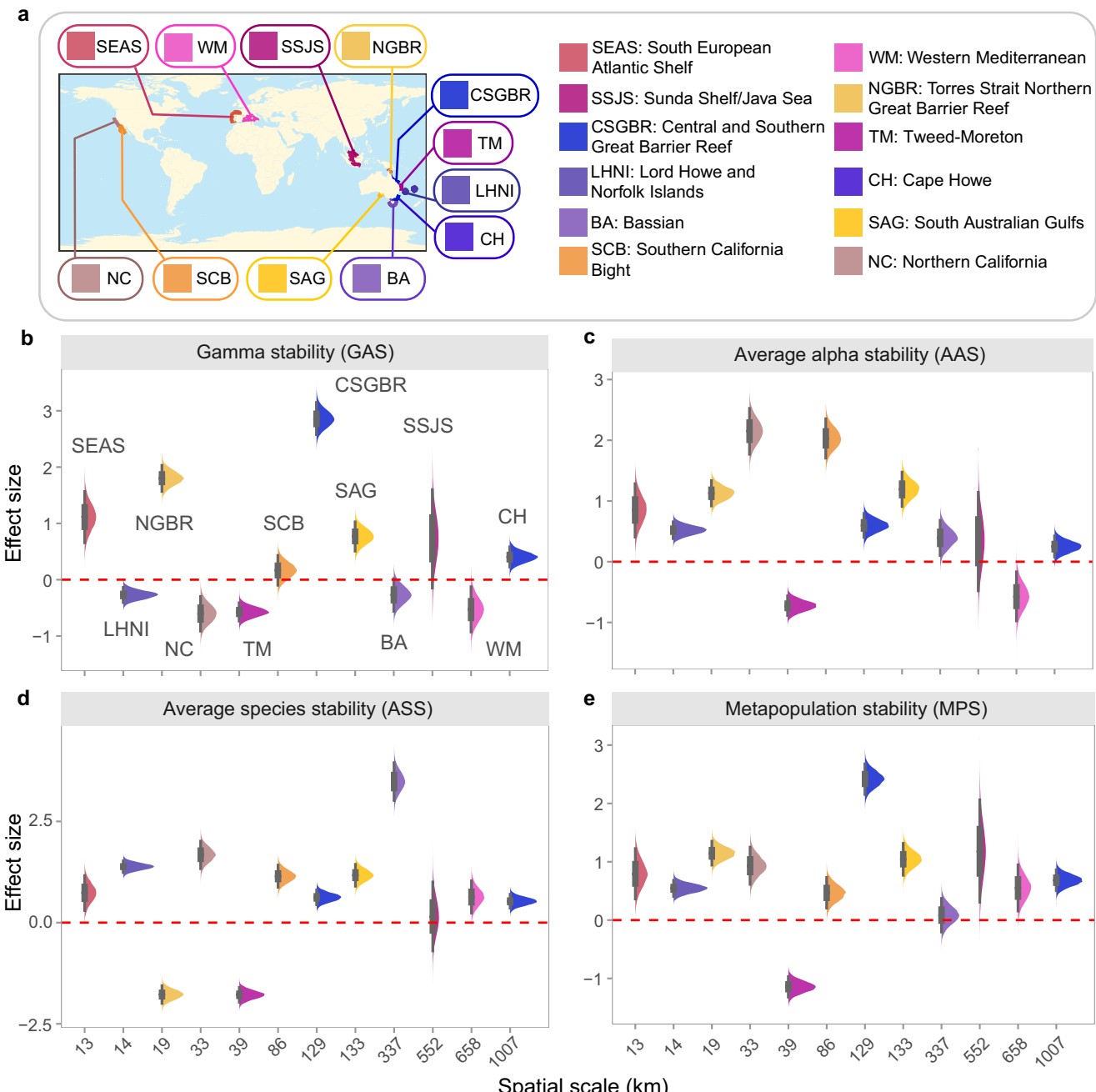

**Fig. 5 | Stability of marine protected area (MPA) networks. a** Ecoregions used in the analysis of stability and asynchrony at the metacommunity level. There were at least two MPA and two open area sites in each ecoregion (see Supplementary Table 5 for details). Panels **b**–**e** show the posterior distributions of effect sizes comparing different stability measures between MPAs and open areas in relation to the spatial scale of the MPA network (see also Fig. 1). See Supplementary Fig. 12 for asynchrony. Distributions are shown with 66% (thick bar) and 95% (thin bar) uncertainty intervals ($n = 12,000$ posterior samples). Intervals that do not overlap with 0 (dashed red line) are considered significant. Positive (negative) effect sizes indicate larger (lower) stability in MPAs than in open areas.

species contributing less and less to the group STI with increasing latitude. In contrast, a positive trend of abundance with marine heatwaves for trophic groups with a STI below the threshold could indicate a transient stage where warm-water species have not yet established, and native species with low STI values are resisting intensifying marine heatwaves.

In contrast to grazers, carnivores required a STI value above the threshold to maintain a positive trend of abundance at intermediate to high intensity of marine heatwaves and this occurred only in MPAs. Fish carnivores include many large body-sized species with relatively low thermal tolerance, which generally decreases with body size and trophic position due to high metabolic demands arising from foraging activity[64]. One mechanistic explanation for the negative relationship between body size and thermal tolerance is that warming enhances metabolic rates and large organisms may be more thermally limited than smaller ones owing to physiological constraints (e.g., oxygen limitation)[64]. Carnivores are also a major target of commercial and recreational fishing. Accordingly, carnivores showed a more pronounced peak in abundance in MPAs than in open areas. Whether these trends were driven by increased dominance of extant thermally resistant species, expansion of warm-water species, or a combination of both, will require further analysis.

Remoteness influences various aspects of reef fish ecology. For example, proximity to human populations was associated with

reduced biomass of reef fishes, smaller sizes of individuals, and fewer species[40]. Distance from direct human disturbance was also a key feature in maximizing conservation benefits of MPAs[8,65]. Our results emphasized the joint effects of remoteness and protection to enhance the stability of reef fish communities. Specifically, remoteness and functional richness were the primary pathways promoting species stability in open areas. Yet, remoteness can also increase the strength of ecological dependences and specializations, such as between fishes and corals, and thus increase the vulnerability of remote reefs to species loss through cascading effects across networks of interacting species[41].

Although numerous studies have documented the positive effects of individual MPAs on biodiversity, whether these findings also apply to population and community resilience across entire MPA networks has remained an open issue[66], as has the extent to which spatial scale affects conservation outcomes. Criteria have been proposed to design MPA networks that can address multiple conservation benefits, including increasing resilience to climate change[44,45]. The location and separation of individual MPAs is important for determining spatial and thermal refuges for vulnerable species and ensuring genetic flow and exchange of individuals through network connectivity[15,44,45]. Well-connected MPAs can benefit from the exchange of individuals that help mitigate local impacts and the effects of climate change[15,17,24]. Although knowledge of the dispersal capabilities of many reef fishes is limited, increasing evidence suggests that long-distance dispersal (10 s to 100 s of km) may be more common than currently thought[52]. Our analysis of MPA networks supported this view. Using graph theory, we found that geographic isolation did not preclude biological connectivity, suggesting that even the most isolated sites could be part of a metacommunity network.

Large MPA networks are expected to include a greater breadth of key habitats and environmental conditions, potentially increasing the portfolio of responses against climate uncertainties[45]. Thus, we hypothesized that gamma stability and the underlying stability and asynchrony mechanisms would increase with the spatial scale of MPA networks. Counter to our expectation, we found no relationship linking stability and asynchrony to spatial scale, number of MPAs and number of sites in metacommunities. Nevertheless, metapopulation stability and, to a lesser extent, average alpha and species stability were consistently greater in MPAs than in open areas. These results support the hypothesis that MPAs can promote gamma stability by mitigating population and species fluctuations, suggesting that even small MPA networks (sites 1-10 km apart) can provide conservation benefits to fish communities.

In conclusion, we provide strong evidence that the benefits of well-enforced MPAs extended beyond the direct effects of mitigating human disturbances. By fostering species abundance and stability, maintaining asynchronous fluctuations and preserving functional richness, MPAs can help stabilize reef fish communities to abrupt changes in climate such as those associated with marine heatwaves. Although reef fishes will be increasingly challenged by the cumulative effects of human pressures and global change in the next decades, they generally have greater margins of adaptation and resilience to marine heatwaves if released from direct human disturbances. As such, MPAs have the potential to play an increasingly important role in promoting reef fish stability in a warming ocean.

## Methods
### Reef fish timeseries
All analyses were performed in R 4.1.3[67]. We assembled timeseries of reef fish abundance from two globally distributed databases, Reef Life Survey (RLS, https://reeflifesurvey.com/) and Reef Check (RC, https://www.reefcheck.org/), published datasets[68,69] and scientific monitoring programs (Supplementary Table 7). All data consisted of quantitative surveys of reef fish abundances obtained by a combination of marine

scientists and trained recreational SCUBA divers, using standardized visual methods. Methodological details, data curation and diver training are provided in refs. 70,71 for RLS and ref. 72 for RC. All surveys were conducted along transects, with the exception of data from ref. 69, which were obtained from 15-m diameter cylindrical plots. Data were aggregated by site and year by summing the abundance of individual fish species across replicate transects (and cylindrical plots), when present. We retained sites with at least five years of observation, which is appropriate for the analysis of stability and population trends in a wide range of taxa[28,50,73]. The final dataset consisted of 71,269 timeseries of population abundances from 2269 reef fish species sampled in 357 MPA and 747 open area sites across 50 Marine Ecoregions. Timeseries ranged from 5 to 28 years between 1992 and 2021 (February).

We identified well-enforced MPAs using the criteria set by the International Union for Conservation of Nature (IUCN)[74], as areas classified either as "No-Take All" or falling in the I-III categories of protection. Expert opinions from data providers and information from published studies were used to determine the level of enforcement when IUCN categories were not applicable or the "No-Take" status was not reported. For example, Medes Islands in the Mediterranean have IUCN category V and has no reported "No-Take" status, but it is typically considered a well-enforced MPA[75]. Similarly, sites included in the first zoning plan of the Great Barrier Reef Park (GBRP), which was established in 1981, have neither "No-Take" status reported nor IUCN category applicable. Co-author ME distinguished between MPAs and open areas in the dataset provided for both the first and the second zoning plan, which was established in 2004. Expert opinion matched IUCN criteria for the second zoning plan, as MPA sites correspond to IUCN category II and "No-Take-All" status.

### Environmental data
We considered two environmental variables as putative drivers of stability and asynchrony: marine heatwaves as an indicator of thermal stress and remoteness (the travel time to large cities) as a proxy measure of direct human pressure. Marine heatwaves were identified from daily Sea-Surface-Temperatures (SST) using the National Oceanic and Atmospheric Administration (NOAA) daily optimum interpolation gridded dataset V2.1 in the period 1 January 1982 to 31 December 2020[76]. The dataset is a blend of observations from satellites, ships, and buoys and includes bias adjustment of satellite and ship observations to compensate for platform differences and sensor biases. Remotely sensed SSTs were obtained through the Advanced Very-High-Resolution Radiometer and interpolated daily onto a 0.25° x 0.25° spatial grid globally. Data were downloaded in January 2022 from https://www.ncei.noaa.gov/data/sea-surface-temperature-optimum-interpolation/v2.1/access/avhrr/.

A marine heatwave can be defined as an anomalously warm water event with daily SSTs exceeding the seasonally varying 90th percentile (climatological threshold) for at least 5 consecutive days[36,37]. The climatology was derived from the 30-yr period 1982 to 2011. We used this period as our baseline to identify marine heatwaves to comply with the recommendation of using at least 30 years for deriving a climatology, while limiting the number of instances in which the climatology extended beyond the year in which a marine heatwave was identified[36]. This occurred at 40 of the 1104 sites used in the alpha stability analysis and involved less than 2% of the 46,976 marine heatwave events identified in the study. Removing these sites from the analysis did not change the results (Supplementary Figure 15). The climatological mean and threshold were computed for each calendar day within a 11-day window centered on the focal calendar day across all years within the climatological period. The mean and threshold were further smoothed by applying a 31-day moving average. Two events with a break of less than 3 days were considered the same marine heatwave. All marine heatwave events were computed relative to the threshold – i.e. as the

difference between the observed SST and the threshold SST. Characteristic measures of marine heatwaves, including mean, maximum, and cumulative intensity, were obtained by pairing events with fish timeseries at the site level. That is, marine heatwaves characteristics were aggregated over the same years in which fishes were sampled. We used mean marine heatwave intensity as the primary metric to quantify marine heatwaves, but we also performed a sensitivity analysis based on cumulative intensity (Supplementary Fig. 4), whereas maximum marine heatwave intensity was used to define thermal thresholds at individual sites. Marine heatwaves were identified and analyzed with the R package heatwaveR[77], using SST timeseries with less than 10% of missing data.

We quantified the "remoteness" of each site as the travel time (in hours) to the closest major city (>100,000 residents), using the procedure first developed for terrestrial environments by Weiss et al.[78] and adapted to marine localities by Strona et al.[41] Briefly, travel time was computed from a global friction surface map (at the resolution of 1 km$^2$) indicating the average speed at which humans can travel through each pixel using the fastest possible aquatic and terrestrial means (thus excluding aerial transportation) and then applying an algorithm to identify the least-cost path (i.e. the shortest travel time) from each site to the closest major city[41].

## Functional richness

Prior to analysis, species names were matched with the World Register of Marine Species[79] and the FishBase[80] database for validation, accessed through the R packages *worrms*[81] and *rfishbase*[82], respectively. We compiled six traits for each of the 2,269 fish species in the dataset, representing body size, trophic position, gregariousness, water position, intrinsic vulnerability to extinction and thermal affinity. These traits covered attributes determining species life history, trophic ecology, habitat preferences, behavior and species temperature distribution[71,83]. Gregariousness and water position were ordered variables, the first coded as solitary, pairing or schooling categories and the second coded as benthic (sedentary), demersal (swimming near the bottom), pelagic-reef (swimming away from the bottom within a reef) and pelagic (swimming away from the bottom among reefs) categories. The other traits were continuous variables: body size, reflecting the theoretical maximum size attainable by a species based on its growth curve; trophic position, describing the position of each species in the food web; intrinsic vulnerability, a synthetic index of the likelihood of a species to go extinct in response to fishing. Finally, thermal affinity, quantified through the Species Temperature Index (STI), measured the upper realized thermal niche of each species. This analysis required matching spatial information of species occurrences with long-term SST means. We obtained species occurrences from the Ocean Biodiversity Information System (OBIS: https://obis.org/) using the R package *robis*[84] and SST long-term means for each occurrence location from the Bio-ORACLE v2.0 database[85]. To remove possible outliers, we first pruned the occurrence data by excluding extreme SST – i.e. values below the fifth and above the 95$^{th}$ percentiles of the temperature distribution occupied by each species. We then calculated the upper realized STI as the 95$^{th}$ percentile of the pruned temperature distribution of each species. All other traits were obtained from FishBase[80]. Continuous traits were averaged at the genus or family level for the fishes that could not be resolved at the species level (8% of the taxa); for ordinal traits, we first determined the most frequent attribute across all members within a genus or family and then converted this trait into the corresponding ordinal score.

Several functional diversity measures can be computed from a species by trait matrix[33]. We used the function *alpha.fd.multidim* in R package mFD[86]. The key step of the analysis is the construction of a multidimensional trait space, which is usually done through a Principal Coordinate Analysis (PCoA) applied to a Gower similarity matrix of the original species by trait matrix. Gower similarity can handle categorical, ordinal and continuous traits with missing data simultaneously, which is a desirable property for fish traits, which typically include variables of different nature, as in our analysis[87]. PCoA axes define a reduced multidimensional trait space within which several indices of functional community structure can be obtained at the site scale. We used the first three PCoAs in our analysis, which explained 40% of the variance on average, as a compromise between quality of trait space representation and computational speed. We focused primarily on functional richness, the proportion of the multidimensional trait space filled by all species in a site, as this measure was independent of other functional indices (correlation coefficients and 95% confidence intervals – functional richness vs. functional diversity: 0.066, 0.007–0.12; functional richness vs. functional evenness: 0.018, −0.04-0.08; functional richness vs. functional dispersion: 0.09, 0.03–0.15), which were significantly correlated (correlation coefficients and 95% confidence intervals – functional diversity vs. functional evenness: 0.15, 0.09–0.21; functional diversity vs. functional dispersion: 0.54, 0.50–0.58; functional evenness vs. functional dispersion: 0.31, 0.26–0.37). Furthermore, these alternative indices performed less well than functional richness in SEMs (see *Methods: Sensitivity analyses and null models*).

## Stability and asynchrony

We computed six measures of stability: alpha and species stability at the site scale and gamma stability (GAS), average alpha stability (AAS), average species stability (ASS) and metapopulation stability (MPS) at the metacommunity scale[5,27,32,50,88] (within ecoregions).

Alpha stability at the site scale was simply the inverse of the coefficient of variation of total fish abundance at a site:

$$\alpha_{Stab,i} = \frac{\mu_i}{\sigma_i} \quad (1)$$

where $\mu_i$ and $\sigma_i$ are the temporal mean and standard deviation of total fish abundance at site $i$, respectively.

Species stability at the site scale was the mean stability among species weighted by relative species abundance:

$$SP_{Stab,i} = \left( \sum_{j(i)} \frac{\mu_{j(i)}}{\mu_i} \frac{\sigma_{j(i)}}{\mu_{j(i)}} \right)^{-1} \quad (2)$$

where $\mu_{j(i)}$ and $\sigma_{j(i)}$ are the temporal mean and standard deviation of abundance of species $j$ at site $i$, respectively.

Gamma stability was obtained as:

$$GAS = \frac{\mu_M}{\sigma_M} \quad (3)$$

where $\mu_M$ and $\sigma_M$ are the temporal mean and standard deviation of total fish abundance in metacommunity $M$.

Average alpha stability in a metacommunity was calculated as the sum of the stability values of individual sites, weighted by relative site abundance in the metacommunity:

$$AAS = \left( \sum_i \frac{\mu_i}{\mu_M} \frac{\sigma_i}{\mu_i} \right)^{-1} \quad (4)$$

Average species stability was obtained by summing the weighted stability of individual species in a site (from Eq. (2)) over sites and

weighting by site relative abundance in the metacommunity:

$$ASS = \left( \sum_i \frac{\mu_i}{\mu_M} SP_{Stab,i} \right)^{-1} \quad (5)$$

where $\mu_{j(i)}$ and $\sigma_{j(i)}$ are the temporal mean and standard deviation of abundance of species $j$ at site $i$.

Finally, metapopulation stability was computed as the sum of the stability of total population abundance for each species in the metacommunity, weighted by relative species abundance:

$$MPS = \left( \sum_j \frac{\mu_j}{\mu_M} \frac{\sigma_j}{\mu_j} \right)^{-1} \quad (6)$$

where $\mu_j$ and $\sigma_j$ are the temporal mean and standard deviation of total abundance of species $j$ in the metacommunity.

We computed five measures of asynchrony: species asynchrony at the site scale and spatial community asynchrony (SCA), spatial species asynchrony (SSA), average species asynchrony (ASA) and metapopulation asynchrony (MPAS) at the metacommunity scale. We first quantified synchrony using both *Gross* and *Loreau and de Mazancourt* (LdM) measures[27,46,50,89] and then converted these measures into asynchrony by changing sign (Gross) or by subtracting synchrony from unity (LdM). Gross et al.[46] quantified the average synchrony among species in a community as the mean correlation coefficient between the temporal abundance of each species vs. the temporal vector of the total abundance of all the other species. The index varies between −1 and 1, reflecting maximum synchrony and asynchrony, respectively, after changing sign. We used the modified version of Gross index that weights correlation coefficients by relative species abundance[50]:

$$\eta_i = - \sum_j \left[ \frac{\mu_{j(i)}}{\mu_i} r \left( \boldsymbol{A}_{j(i)}, \sum_{k \neq j} \boldsymbol{A}_{k(i)} \right) \right] \quad (7)$$

where $\eta_i$ is the weighted asynchrony index at site $i$ and the term $r(\boldsymbol{A}_{j(i)}, \sum_{k \neq j} \boldsymbol{A}_{k(i)})$ indicates Pearson's $r$ correlation between the temporal vector of abundances of species $j$ in site $i$ ($\boldsymbol{A}_{j(i)}$) and the vector originating from the sum of the abundances of all the remaining $k$ species in the community ($\boldsymbol{A}_{k(i)}$). Spatial community asynchrony quantified the average dissimilarity of temporal fluctuations among sites in a metacommunity, weighted by relative site abundance:

$$SCA = - \sum_i \left[ \frac{\mu_i}{\mu} r \left( \boldsymbol{A}_i, \sum_{m \neq i} \boldsymbol{A}_m \right) \right] \quad (8)$$

where $\boldsymbol{A}_i$ is the temporal vector of total community abundance at site $i$ and $\sum_{m \neq i} \boldsymbol{A}_m$ is the temporal vector originating from the sum of the abundances over the remaining $m$ sites in the metacommunity. Following the same rationale, spatial species asynchrony quantified the average dissimilarity of temporal fluctuations among populations in the metacommunity, weighted by relative species abundance:

$$SSA = - \sum_i \sum_j \left[ \frac{\mu_{j(i)}}{\mu_i} \frac{\mu_i}{\mu} r \left( \boldsymbol{A}_{j(i)}, \sum_{m \neq i} \boldsymbol{A}_{j(m)} \right) \right] \quad (9)$$

were $\boldsymbol{A}_{j(i)}$ is the temporal vector of the abundance of species $j$ in site $i$ and $\sum_{m \neq i} \boldsymbol{A}_{j(m)}$ is the temporal vector of total fish abundance summed over the remaining $m$ populations of species $j$ in the metacommunity. Average species asynchrony quantified the average dissimilarity of temporal fluctuations among all species in a site (Eq. (7)) averaged among sites and weighted by relative site abundance in the metacommunity:

$$ASA = - \sum_i \left( \frac{\mu_i}{\mu} \eta_i \right) \quad (10)$$

Finally, metapopulation asynchrony quantified the dissimilarity in temporal fluctuations among total species abundances, weighted by relative species abundance in the metacommunity:

$$MPAS = - \sum_j \left[ \frac{\mu_j}{\mu} r \left( \boldsymbol{A}_j, \sum_{k \neq j} \boldsymbol{A}_k \right) \right] \quad (11)$$

where $\boldsymbol{A}_j$ is the temporal vector of the total abundance of species $j$ in the metacommunity and $\sum_{k \neq j} \boldsymbol{A}_k$ is the temporal vector of total fish abundance summed over the remaining $k$ species in the metacommunity.

For comparative purposes we recalculated all asynchrony measures from 1- $\varphi$, with $\varphi$ indicating LdM synchrony:

$$\varphi = \frac{\sigma^2}{\left( \sum_j \sigma_j \right)^2} \quad (12)$$

where $\sigma^2$ is the variance in total fish community abundance and $\sigma_j$ is the temporal standard deviation of abundance of species $j$. Equation (12) can be modified to quantify asynchrony at all the hierarchical levels addressed in Eqs. (7)–(11) (see also ref. 32). Weighted Gross asynchrony was computed using a custom function, whereas LdM asynchrony was computed using function *synchrony* in the R package *codyn*[90].

## Data analysis

**Community-level analysis.** We used Linear Mixed Effect Models to examine the relations between alpha stability (Eq. (1)), species stability (Eq. (2)), species asynchrony (Eq. (7), functional richness and their putative drivers (marine heatwaves and remoteness) and to fit piecewise Structural Equation Models (SEMs)[49]. All models included a random intercept for study ID, which coded for the different data sources (Supplementary Table 8) and accounted for possible generic differences in methodology among monitoring programs. In addition, we explicitly controlled for sampling effort by including the total area sampled at each site in each year as an offset in all models (see section below, *Controlling for sampling effort*, for details). All variables were standardized to z-scores (scaled and centered over the entire dataset) prior to analysis to provide a common scale for both responses and predictors; stability measures, remoteness and sampled area were log-transformed before standardization to improve normality. We first examined separate relationships for MPAs and open areas to match the models used in SEM, but also tested for interactions between predictors of stability, asynchrony species and functional richness and level of protection. The adequacy of model fits was assessed through a variety of diagnostics, based primarily on visual assessment of residuals using the R package *performance*[91].

SEMs were generated separately for MPAs and open areas to reflect the hypothesized direct and indirect casual pathways among alpha and species stability, species asynchrony, functional richness, marine heatwave mean intensity and remoteness. We fitted individual pathways using the same model structure and variable transformations employed in mixed-effect models. Marine heatwaves and remoteness were exogenous variables in all models, whereas alpha stability was only an endogenous variable. All other predictors were both endogenous and exogenous variables. We started by fitting nearly-saturated global models where each endogenous variable included paths from all exogenous variables in addition to the remaining

endogenous variables, but avoiding reciprocal paths between the same variables. The only exception was the relationship between species stability and asynchrony, which was not considered since we had no a priori hypothesis about the direction of a causal path between these variables. Thus, functional richness was initially modeled as a function of marine heatwaves and remoteness; the models for species stability and asynchrony included functional richness and its predictors (marine heatwaves and remoteness); alpha stability was modeled as a function of all the other variables. We used Fisher's $C$ statistic to evaluate the adequacy of the global models to reproduce the hypothesized causal paths[49]. A model can be considered adequate when the $C$ statistic is not significant ($p > 0.05$). The initial model for open areas was properly specified (Fisher's $C = 0.1$, 2 d.f., $p > 0.05$), whereas the MPA model was not (Fisher's $C = 7.7$, 2 d.f., $p < 0.05$). Removing the not significant link ($p > 0.5$) from remoteness to functional richness improved the MPA model making Fisher's $C$ not significant ($C = 8.65$, 4 d.f., $p > 0.05$). Results are shown as standardized effect sizes; direct and indirect effects (Fig. 3c, Supplementary Fig. 5) were extracted from SEMs using function *semEff* from the same R package[92]. Confidence intervals for standardized effects sizes were derived by nonparametric bootstrap of the fitted modes using function *bootEff* in package *semEff*.

We modeled thermal sensitivity trends using Generalized Additive Mixed Models (GAMMs) to account for the non-linear relationships between MWHs and the abundance of the four fish trophic categories. GAMMs included a tensor smooth term of marine heatwaves in interaction with MPA and open area conditions and a random smooth term for study ID. The main effect of MPA vs. open areas was evaluated in the linear part of the model. Assumptions were assessed visually by evaluating the distribution of model residuals, plots of residuals vs. fitted values and the linear predictor and plots of deviance residuals vs. theoretical quantiles. GAMMs were fitted using function *gam* in R package mgcv[93].

**Controlling for sampling effort.** Our analysis required controlling for sampling effort. Although sampling methods of reef fish abundance were consistent within individual survey programs, the total area sampled varied among sites due to differences in the number of replicates and, for different programs, in the size of individual transects[94]. One way to account for sampling effort when investigating population trends is to divide abundance (counts) by sampled area and analyze the resulting density estimates. Unfortunately, this was not a viable approach for our analysis of stability and asynchrony because sampled area was a constant at any given site and dividing fish abundances (or any other variable) by a constant results in exactly the same values of stability and asynchrony as those obtained analyzing the original data. That the coefficient of variation – from which our measures of stability are derived – does not change when the input data are multiplied by a constant, is a well-known property of this statistic[95]. The same applies to measures of asynchrony since dividing timeseries of fish abundances by a constant leaves the relative differences among timeseries unchanged.

An alternative way to control for sampling effort is to include an offset in the model[96]. An offset is a fixed quantity associated with each observation that is used to scale the response variable, such that its influence is accounted for in the model. The offset is added to the linear predictor with a fixed coefficient of 1 (i.e. no regression coefficient is estimated for an offset) and the scaling is simply achieved by subtracting the offset from the response variable. When both the response and the offset variables are log-transformed, the scaled response variable becomes a log-response ratio (since the difference between two log-transformed quantities is equivalent to the logarithm of their ratio), which is the typical use of an offset in Poisson or binomial regression to model rates or proportions. Nevertheless, offsets can be included in other types of regression models and they are commonly employed in studies that combine data from multiple programs with varying levels of sampling effort, such as in bird surveys[96,97].

Scaling the response variable by the offset requires that both variables are on the same scale. This was achieved by standardizing (i.e. scaling and centering) the response and the predictor variables, including the offset (sampled area), to z-scores. Thus, our community-level analysis shows fitted trends for scaled variables obtained as the difference between each response variable and the offset, after standardization.

Indeed, an offset may not be needed in linear models, where one could simply work with log-response ratios[98]. We show this equivalence in Supplementary Fig. 5, where the whole analysis is repeated by dividing each response variable by sampling effort (both log-transformed) and removing the offset from the linear model. This analysis does not assign any fixed coefficient to sampling effort, since it is now part of the response variable. Results are very similar to those obtained with an offset (e.g., the stronger negative relation between stability and marine heatwave intensity in open areas compared to MPAs). These outcomes reassure that our analysis is robust to specific choices of data transformation and that similar results are obtained whether scaled response variables are expressed as log-response ratios or as differences between standardized variables through the offset. We opted to present results based on the offset in the main text, since this improved data visualization compared to log-response ratios (compare Fig. 2b–k and Supplementary Fig. 1 with Supplementary Fig. 5).

**Metacommunity networks and connectivity.** We derived minimum spanning tree graphs (networks) from geographic (least-coast path distance by the sea) and biological (using Jaccard dissimilarity) distances for each metacommunity. A minimum spanning tree includes the minimum number of shortest distances to maintain all sites (nodes) connected without closed paths among nodes[54]. We then computed degree and closeness centrality to characterize the topology of each metacommunity network and to investigate the relationships between geographic and biological connectivity. Specifically, we employed least-squares linear regression to relate closeness centrality measured on a geographically-derived graph to degree centrality measured on a biologically-derived graph. We used functions *graph.adjacency*, *mst*, *strength* and *closeness* from package igraph[99] to generate networks from distance matrices, derive minimum spanning trees and to calculate degree and closeness centrality, respectively. Degree and closeness centrality were weighted by 1/distance and scaled before analysis. Jaccard dissimilarity was computed using function *vegdist* in package vegan[100].

**Metacommunity stability and asynchrony.** We compared metacommunity stability and asynchrony between MPAs and open areas within ecoregions. First, we selected ecoregions that had at least two MPA and two open area sites sampled simultaneously for at least five years. This was necessary to obtain comparable stability and asynchrony measures. There were 12 ecoregions that met these criteria. Since there were many possible ways to combine sites and years, we developed an algorithm to select the combination of years that maximized the number of MPA and open area sites (an alternative algorithm that maximized the length of matching timeseries yielded too few sites in most ecoregions).

Second, we calculated a matrix of least-cost path distances by the sea (i.e. avoiding land masses) among the selected sites for each of the 12 metacommunities using function *costDistance* from *terra* package in R[101]. We used these distance matrices to match MPA and open area sites within the spatial scale defined by the maximum distance separating any two MPA sites within an ecoregion. For each MPA site we identified all other MPA and open area sites within the defined spatial scale and computed all metacommunity stability and asynchrony

measures from these sites. This procedure was repeated for all MPAs in a metacommunity and the results were averaged. Inevitably, all MPA sites became selected at each iteration (they were all included within their maximum distance, by definition), thus, only the stability and asynchrony measures obtained from one iteration were retained for MPAs. As a sensitivity test, we repeated the analysis by matching MPA and open area sites within a spatial scale of 50-100 km, which was intermediate between the maximum distances separating MPAs in metacommunities (Supplementary Table 5), with 100 km representing a potential upper limit of direct fish dispersal[40,56]. Although our matching procedure used the same sites more than once, averages were independent between MPAs and open areas.

Third, we compared the stability and asynchrony measures between MPAs and open areas within each of the spatial scales defined above. To do so, we developed a simulation approach to obtain robust estimates of variances for each stability and asynchrony measure and level of protection. Although we had access to primary fish abundance data, Eqs. (2)–(6) and Eqs. (8)–(11) necessarily generated a single value for each metacommunity precluding the direct estimation of variances. We addressed this problem through a Jacknife (leave-one-out) simulation approach, which consisted in recalculating all stability and asynchrony measures for each metacommunity by excluding one species at the time. The resulting variances were used to derive the *Hedge's g* effect size of the difference between MPAs and open areas for each measure, which we analyzed in a Bayesian meta-analytical framework. We used a model of the following form:

$$y_m = gaussian(\theta_m, \sigma_m^2) \tag{13a}$$

$$\theta_m = gaussian(\mu, \tau) \tag{13b}$$

where $y_m$ was the estimated *Hedge's g* effect size for any of the measures analyzed in metacommunity $m$, which was assumed to originate from a Normal distribution centered on the true effect size $\theta_m$ with variance $\sigma_m^2$. Metacommunity $m$ was considered a random sample from a population of possible metacommunities, such that $\theta_m$ itself originated from a Normal distribution with true mean $\mu$ (the true population-level effect size) and dispersion parameter $\tau$. We used weakly informative priors for parameters (a normal priori for $\mu$ and a Cauchy prior for $\tau$):

$$Pr(\mu) = N(0, 1) \tag{13c}$$

$$Pr(\tau) = Cauchy(0, 1) \tag{13d}$$

Separate models were fitted for each metacommunity stability and asynchrony measure using function *brm* from the *brms* R package[102]. Models run for 4000 iterations, 1000 burn-in iterations and 4 chains; other tunable parameters in *brm* function were left to their default value. Model convergence was assessed through visual inspection of trace plots and ensuring that the R̂ parameter – a key diagnostic of convergence – was equal to unity. Since gamma stability was examined within metacommunities, which generally included data from individual programs with consistent methods and sampling effort, an offset was not included in these analyses. Finally, we examined the relations between posterior distributions and three attributes of MPA networks, spatial scale, number of MPAs and number of sampled sites, through linear regression.

### Sensitivity analyses and null models
We performed a series of additional tests to evaluate the sensitivity of our results to analytical detail and methodological differences among monitoring programs. Checks were particularly needed for community-level analyses, which compared data across monitoring programs. We assessed the robustness of results and conclusions from the analysis of alpha stability to specific choices of asynchrony (Gross vs. LdM) and functional measures (richness, diversity, evenness, dispersion) and to detrending of timeseries[47]. All functional measures were weakly associated with alpha stability and asynchrony, but functional richness had stronger path coefficients than the other functional measures in SEMs and resulted in a lower Fisher's *C* score contributing to a better representation of the hypothesized casual pathways.

Monitoring programs differed in their taxonomic scope: although most of them were designed to survey all the species occurring in sampling units, some targeted a pre-determined subset of the species (e.g., RC). We performed two analyses to evaluate whether differences in taxonomic scope and other methodological details among monitoring programs affected the main results. First, we evaluated the robustness of one key result, the positive effect of MPAs on alpha and species stability, asynchrony and functional richness with intensifying marine heatwaves, by excluding study IDs with 50 species or less (Supplementary Fig. 6). Second, we used sample coverage[48] to evaluate whether fish communities were adequately sampled in MPAs and open areas, regardless of differences in taxonomic scope, sampling effort and size of sampling units (transects, cylindrical plots) among monitoring programs. Sample coverage is a measure of sample completeness and gives the proportion of the total number of individuals in the community that belong to the species represented in a sample of that community. Sample coverage can be calculated by rarefying (subsampling) the community, or by extrapolating abundance or incidence data to a pre-determined value (typically, twice the total observed abundance or number of samples)[48]. Subtracting sampling coverage from unit gives the "coverage deficit", the probability that a newly added individual (for abundance data) or sampling unit (for incidence data) belongs to a previously unseen species in the sample. We compared sample coverage between MPAs and open areas for different size categories of sampling units (transects or cylindrical plots) using function iNEXT in the same R package[103].

Although most programs started after the enforcement of protection, some (e.g., the GBRP) embraced both before and after periods. These timeseries could include spikes of fish abundance and diversity in response to protection that may not have occurred in timeseries including only after data, with unknown consequences on estimates of stability and asynchrony. To assess this potential bias, we repeated the analysis of alpha stability by excluding data sampled before the establishment of an MPA from those timeseries that encompassed both periods. Results were qualitatively similar to those of the main analysis and are thus not reported here.

Finally, we ran null models to assess whether fish species fluctuated more or less asynchronously than expected by chance in metacommunities. Null models consisted of 999 iterations of the cyclic shift algorithm, a common method to preserve temporal autocorrelation in simulated timeseries. We applied the cyclic shift algorithm independently to each individual species at a site. Observed timeseries were considered one realization of the null model and were combined with those originated by the cyclic shift algorithm, using the function *cyclic_shift* in R package codyn[90].

### Reporting summary
Further information on research design is available in the Nature Portfolio Reporting Summary linked to this article.

### Data availability
All the data required to reproduce the results of this study have been deposited in the Figshare database under accession code https://figshare.com/s/ffa4f5cb22799532bbc1 and on Github at https://github.com/bencecc/ReefFishStability[104]. Global SST data can be accessed at

https://www.ncei.noaa.gov/data/sea-surface-temperature-optimum-interpolation/v2.1/access/avhrr/. Fish abundance data were obtained from: Reef Life Survey (https://reeflifesurvey.com/), Reef Check (https://www.reefcheck.org/), BioTime (https://onlinelibrary.wiley.com/doi/10.1111/geb.12729), the Long-Term Monitoring of Coral Reef Fish Assemblages in the Western Pacific (https://www.nature.com/articles/sdata2017176). Additional data were provided by co-authors D.J.K., D.C.R., M.J.E., B.H.C., E.J.G., N.S.B., G.J.E., J.A.G.C., E.A,. B.H.

## Code availability
Analyses were run in R version 4.1.3. The code used in this study is available on GitHub[104].

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

## Acknowledgements

We thank the Citizen Sampling Programs Reef Life Survey and Reef Check for contributing part of the data used in this study. Data from Reef Life Survey are managed through, and were sourced from, Australia's Integrated Marine Observing System (IMOS) – IMOS is enabled by the National Collaborative Research Infrastructure Strategy (NCRIS). LBC acknowledges the contribution by the European Union's Horizon 2023 Research and Innovation Program under grant agreement No. 101060072 ACTNOW and by the University of Pisa grant PRA_2020_76. BH was supported by the Monitoring Program of the Catalan Natural Parks funded by the Generalitat de Catalunya. BHC was supported by national funds through FCT - Foundation for Science and Technology,

I.P. (Portugal), in agreement with the University of Algarve, in the scope of Norma Transitória with the research contract DL57/2016/CP1361/CT0038 and through the CCMAR strategic projects UIDB/04326/2020, UIDP/04326/2020 and LA/P/0101/2020. EJG was supported by FCT - Foundation for Science and Technology funds through the programs UIDB_P/MAR/04292/2020 (MARE-ISPA). DCR was supported by the U.S. National Science Foundation through the Santa Barbara Coastal Long Term Ecological Research project (OCE 1831937). DJK was supported by the United States National Park Service Inventory and Monitoring Program. FB was supported by the European Union's Horizon 2020 Research and Innovation Program under grant agreement No 869300 FutureMARES and by the University of Pisa grant PRA_2020_76. JAGC is grateful for the funding from the Regional Fisheries and Aquaculture Service — Autonomous Community of Murcia for the regular monitoring of the Cabo de Palos - Islas Hormigas Marine Reserve of Fishing Interest, with the support of the European Maritime and Fisheries Fund. RSS was supported by an Australian Research Council Fellowship FT190100599.

## Author contributions

L.B.C. conceived the study with inputs from A.E.B., D.C.R., F.B., G.S., G.J.E., R.S.S., and B.H.C. L.B.C. performed the analysis and wrote the first draft of the manuscript. D.J.K., D.C.R., M.J.E., B.H.C., E.J.G., N.S.B., G.J.E., J.A.G.C., E.A., B.H. provided data. All authors contributed to the interpretation of the results and to the final version of the manuscript.

## Competing interests

The authors declare no competing interests.
