## [Peer Review File · Nature Communications]

Marine protected areas promote stability of reef fish communities under climate warmingREVIEWER COMMENTS

Reviewer #1 (Remarks to the Author):

General Comments to the authors

This is an interesting study that empirically test if the protection of marine areas offers resilience under heatwaves (MHWs) and human pressures from local to metacommunity scales. To this aim, authors examine a big timeseries database of reef fish species in protected and open sites worldwide. Alpha stability, species stability, species asynchrony and functional richness all declined with increasing mean intensity of MHWs in OAs, but not in MPAs. At the metacommunity scale, there was no clear trend of variation in stability and asynchrony measures across the ecoregions examined. In general terms, the effects at local scale are supported but I have some concerns mainly regarding the metacommunity scale.

1. I think the metacommunity approach is the biggest weakness of this study, and maybe this could be why there was not clear trends at the metacommunity scale.
 - i. First, the flow of individuals is a main determinant of metacommunities stability and resilience of several ecosystems, including marine systems. Moreover, the adaptive capacity to climate change of several ecosystems is strongly affected by the landscape connectivity (or seascape connectivity in this case). Landscape connectivity has been suggested as a climate change adaptation strategy for biodiversity preservation, which could allow species to track changing habitat conditions. In this context, to evaluate resilience of a metacommunity is really important to consider the degree of connectivity of the network.
 - ii. Second, authors assume that distant sites became connected over multiple generations through stepping-stone effects within a metacommunity. This scenario is a full connected network, i.e., all communities are connected among them. However, the metacommunity framework has sense at intermediate connectivity values. Since at low connectivity values, communities are isolated, and at high connectivity value it is a big community.
 - iii. Third, in figure 1 the connectivity among sites is mentioned, but it is not clear how this is considered in the analysis. It is associated to the size (number of communities) and the maximum distance between sites. Why not a SEM for the metacommunity scale? Each metacommunity is an ecoregion? In the discussion (that begins in Line 335) something is slightly mentioned about the location of sites, and the flow individuals through network connectivity. However, is not herein explicitly considered.

Thus, I think it would be great to incorporate analysis that really evaluate the metacommunity scale and improve in the discussion the metacommunity framework.

2. Environmental heterogeneity refers to temporal fluctuations of the metrics used? It could be confused with the variability of an environmental filter in the time, but I think is not the case. It is a bit confuse since is mentioned in the introduction and never more.

3. As it is showed in Figure 3a and 3b, I found interesting that similar direct positive effects were observed in MPAs and OAs (except for the connection between remoteness and alpha stability). Functional richness is positively associated with species stability in protected and non-protected areas. But what promotes functional richness? That is, basic mechanisms that drive ecological relationships independently if it is the MPAs or OAs. The main differences are in the negative effects of MHWs in OAs. So how the results support the hypothesis that MPAs promote reef fish stability at the community level, different from the mechanisms in OAs? And also, why not considered one analysis incorporating the identity of the area (MPAs or OAs). What is the meaning of that well-enforced MPAs that authors said that buffer the impact of MHWs on species and community stability?

4. Remoteness is a proxy of human pressure, measured herein as the travel time to large cities. Which was the range of remoteness used? Was it similar in both areas? In addition, the relationship between remoteness and functional diversity in the MPAs was evaluated? Less functional richness could be expected in more isolated communities.

Minor comments

1. Authors propose several hypotheses about the drivers of resilience in MPAs and OAs in the introduction at local and metacommunity scales. Maybe a table or a conceptual diagram could be incorporated to clarify and synthesize the main hypothesis.
2. Positive and negative effects in the SEM have the same colors that MPAs and OAs.
3. Some details of the analysis about thermal sensitivity trends in Result section, could be moved to methods.
4. Functional richness was defined by three axis of the PCoA, how they were used in the SEM?
5. How open areas are defined? They said that are areas subjected to some form of extractive use.

Reviewer #2 (Remarks to the Author):

It is important to clarify whether Marine Protected Areas (MPAs) is effective to protect biodiversity. Benedetti-Cecchi et al. compiled a large timeseries of population abundance for reef fish species from both protected and open sites world to assess the effects of MPAs to increase stability from populations to metacommunity, particularly in the response to the marine heatwaves. This study shows support that MPAs can increase temporal abundance stability of populations, communities and metacommunities. Overall, I think this study can fill an important gap in assessing the effectiveness of protected areas. However, some parts of the manuscript, particularly the method section, need to be largely clarified and improved.

General comments:

1. Marine heatwaves (MHWs) is an important term/variable to fully understand this manuscript. I think it should be clearly defined in the main text. In the method section, it was still not clear how marine heatwave was calculated. It was stated that the climatology was derived from the 30-yr period 1982 to 2012, but the population timeseries were sampled from 1992 to 2021. Does it mean that it is not necessary the time periods of population trends and marine heatwaves need to be matched in the analyses? The calculation of marine heatwaves needs a threshold, which was stated based on "seasonally varying 90th percentile" daily sea surface temperature (SST). It needs more details, such as how seasons were determined, how the climate time series was used in calculating the threshold, etc. The intensity of marine heatwave was calculated as the difference between the observed SST and the threshold SST. If a given year had multiple marine heatwaves, how the intensity was calculated? If a mean difference between the observed SST and the threshold SST was used and two sites had the same difference, how to distinguish two sites with one and two marine heatwaves events?
2. Figure 4 shows the relationship between abundance MHW intensity for different groups of species. The figure legend describes it shows "temporal trajectories". Did the "abundance" indicate the mean abundance of a group of species across years or the temporal trends of abundance over years? If it indicated absolute abundance (rather than abundance trends), I think the spatial variation in abundance should be determined by many environmental variables, which were not considered presently. Also, I am sure whether the abundance is comparable when different methodologies were used in collected samples.
3. Sampling effort was largely varied across sites in the compiled dataset, as the authors stated. The sampling effort is expected to affect most biodiversity measures. To control for sampling effort, an offset was used in all models. I don't think it was an effective way for models with stability and asynchrony as response variables. In the manuscript, stability was measured as the inverse of coefficient of variation (mean/sd). If the abundance collected is proportional to sampling areas (efforts) across years, both mean and sd will be proportional to sampling areas but the coefficient of variation will not be affected. Asynchrony was measured using correlation between samples (population/species/community) and it will also not be affected by sampling areas if the abundance collected is proportional to sampling areas. I don't say stability and asynchrony are independent of sampling effort because larger sampling areas are expected to have smaller variation over time when heterogeneous parts of a large sampling area can compensate in the fluctuation. However, the sampling efforts cannot be addressed by directly including the sampling

effort as an offset. I think the sampling efforts can be standardized for each population/species/community first and then stability and asynchrony can be calculated based on the standardized data.

4. The manuscript assessed how the effect of MPAs on stability and asynchrony varies related to spatial scale. The authors calculated the distance between pairwise sites in MPAs and used the maximum distance as the spatial scale. If multiple sites are not randomly distributed in MPAs, e.g. five sites distributed closely but one site distributed from the other five sites, the maximum distance can't well represent the distance (spatial scale) among these sites. Also, I don't fully understand this sentence "For each MPA site we identified all other sites within the defined spatial scale and performed all possible pairwise comparisons separately for each level of protection". Because the spatial scale was defined as the maximum distance between MPA sites, all pairwise MPA sites will be within the defined spatial scale. FOR OAs sites, not all sites will be within the defined distance from each MPA site. In the described method, only OAs that were not far from the MPA sites were selected. However, the distances among OAs should be smaller and may be not comparable with the distance among MPA sites.

Specific comments:

Line 46: Fig. 2 showed a positive relationship between stability and remoteness. A higher remoteness means lower human pressure. so, the statement "We find that the intensity of human pressures is positively related to local stability only in protected areas" is incorrect.

Lines 98-102: Does it means high functional richness increases the asynchronous among species and further improves total abundance stability?

lines 131-135: I can't understand how "Thus, by reducing direct human disturbances, MPAs should mitigate population and species fluctuations, increasing gamma stability" can be expected.

Line 178: The subtitle is not easy to understand what will talk about.

Lines 184-186: How to interpret fish abundance increases with MHW in protected areas?

Line 217: "In this section" is not necessary here in the results part.

Lines 254-256: Why effect sizes not deviate significantly from zero can indicate higher gamma stability in OAs than in MPAs?

Line 276: It is not clear that "stronger asynchronous fluctuations" in MPAs.

Lines 288-289: The statement "fishes that experienced warming events beyond their upper realized thermal limit (STI below threshold) increased in abundance maximum intensity of MHWs" was not strongly supported by results (Fig. 4)

Lines 308-313: I don't find these sentences closely connected to the topic discussed in this manuscript.

Lines 443-444: How many species have no species-level trait data?

Lines 490-491: This equation is not absolutely correct. Possible corrections are to write out the $SP[stab, i]$, or remove $sum[j]$.

Line 521: This equation should further multiply $u[i]/u$ in the brace.

Line 555: A typo, "and" rather than "ad".

Line 568: "the model for species asynchrony included species stability": in all results reported, there was no relation between species stability asynchrony. Did the relationship was removed in the model selection?

Line 594: What is the cost layer/raster/data to calculate the least-cost distances?

Line 654: It is necessary to provide more details about the null model to clarify how the model works. Currently, only "cyclic shift algorithm" was given. Is the abundance of a given species in a given site cyclically shifted or does the total abundance in a given site shift?

Figure 2: Why R2 was present in panels b, f, j, k, but not in other panels?

Supplementary Fig. 2: Why R2 was present in panels a, f, i, l, but not in other panels?

Supplementary Fig. 3: Why R2 was present in panels a, b, e, f, but not in other panels?

Reviewer #3 (Remarks to the Author):

This study aggregated fish UVC data from 12 survey programs around the world with the objective of evaluating the stability of assemblages in the face of stressors like warming and human impacts and the degree to which marine reserves can mitigate this. The overall conclusion was that assemblages did seem to be more stable in marine reserves than in areas open to fishing, even mitigating effects of warming and being positively related to fish abundance in reserves. In general I really like the approach taken here, looking at system stability. I think these results would have huge benefits for our understanding of the utility of marine reserves. However I outline a number of issues below, largely to do with unknowns about the data aggregation and its challenges, that could have large impacts on these overall conclusions.

Major comments

I was concerned about how area assessed in each of the various different survey programs was handled when data was aggregated. Certainly these different programs have different areas surveyed, sometimes greatly. And programs like RLS and the AIMS LTMP have different methods to capture different suites of species within their program which have very different survey belt widths. So while I can see the logic in adding areas surveyed for each replicate as a co-variate, it did seem more complicated than needed. Why did you not just convert all values to density and then proceed from there? There is of course an often unrecognized assumption when doing this, namely that abundance scales linearly with area surveyed, but that seems to be a very safe assumption and thus this approach is widely used. So in theory including area as a co-variate should work but here all response variables are standardized first, and area is log transformed prior to this. This would have the effect of decoupling this relationship and I'm guessing you would not get the same answer as had one just used density given the log transform would affect some area values more than others. As this is the basis for everything else that's done in this study, it's a pretty fundamental issue to either clarify or to fix.

Another issue with data aggregation that is not addressed but would seem to be of fundamental importance is the different levels of taxonomic specificity and more importantly, scope, of the different surveys. I'm not familiar with all these survey programs but I do know, for instance, that RLS works to capture cryptic species, while AIMS LTMP does not. However, LTMP does use two different belt widths to capture more versus less mobile taxa very specifically. I'm not familiar with ReefCheck methods but in looking over the information at the link provided in the manuscript, the data captured is mainly at the family level but doesn't seem to include important herbivores like Acanthurids. While I appreciate the focus of the study was not on diversity or richness, these differences in taxonomic coverage would seem likely to influence the potential for stability or asynchrony as I'd imagine both are less likely the more taxa one considers. And these differences in taxonomic coverage would definitely affect the analysis based on total numbers, such as the trophic guild analysis and anything to do with functional richness

The previous two points, in combination with the unexpected result that thermally sensitive species were more common in areas where MHWs were more intense (which tend to be away from the tropics...a point made by the authors) raises concerns as to how much of the patterns observed

in the aggregated data set are just the result of regional differences due to different survey methods (different areas and taxonomic scope). This isn't discussed at all. I see that study ID was included but nowhere are the results of this, or the area coefficient, reported. Similarly, is either stability or asynchrony influenced by the number of taxa (time series) considered? If so, would you expect any latitudinal trends, with fewer species as you move away from the tropics? Can this explain any variation in the data.

The minimum for five years in a data series does seem short though I appreciate there is probably no perfect answer here. Though I was surprised to see the justification for this time period is based on studies in grassland assemblages. Is the scale of temporal dynamics the same as with fish assemblages? That would seem important to clarify in order to justify basing this time period of such a seemingly different system.

I found the acronyms quite confusing as there was no consistent system to them. That said, I did spend 10min trying to come up with a better system and am not confident I did anything more useful. So it's not easily solved but will definitely be an obstacle to the uptake of information here. One observation I'd make is I'm not sure using alpha and gamma helps clarify things. I appreciate those come from diversity standards, but really you could refer to 'site stability' rather than alpha and then 'species stability' makes more sense and both are seen as cutting different ways across the data at the local scale. Then you can have 'average site stability'. Similarly for gamma it could just be meta-community (or MC) stability.

Minor Comments

Ln 551: can you be more clear over what data set the z-scores were generated. I assume the whole data set for each variable, but good to be clear (in case, for instance, it was by regions...which would seem unlikely).

Figure 1. I love the approach of this figure, and it's definitely needed. But I still found it quite confusing after a number of reads. I believe this is because what's shown are time series but really the measures discussed are just derived from the variance and means of these time series in different ways. I wonder if you could restructure this such that you start with time series as in, say, panel c. You then represent the temporal mean and SD on those and then branch off those values in different ways to show how all the other measures are calculated?

We provide a point-by-point response to the comments raised by the reviewers. Please, note that the original comments are in *italic*, whereas our response is in plain text. Where appropriate, we indicate the lines in the manuscript where we have made changes. Please, note that lines refer to the position in the word document with track changes, not in the pdf with accepted modifications.

REVIEWER 1.

General Comments to the authors

This is an interesting study that empirically test if the protection of marine areas offers resilience under heatwaves (MHWs) and human pressures from local to metacommunity scales. To this aim, authors examine a big timeseries database of reef fish species in protected and open sites worldwide. Alpha stability, species stability, species asynchrony and functional richness all declined with increasing mean intensity of MHWs in OAs, but not in MPAs. At the metacommunity scale, there was no clear trend of variation in stability and asynchrony measures across the ecoregions examined. In general terms, the effects at local scale are supported but I have some concerns mainly regarding the metacommunity scale.

We thank the reviewer for the appreciation of our work and for providing constructive comments to the manuscript.

I think the metacommunity approach is the biggest weakness of this study, and maybe this could be why there was not clear trends at the metacommunity scale.

i. First, the flow of individuals is a main determinant of metacommunities stability and resilience of several ecosystems, including marine systems. Moreover, the adaptive capacity to climate change of several ecosystems is strongly affected by the landscape connectivity (or seascape connectivity in this case). Landscape connectivity has been suggested as a climate change adaptation strategy for biodiversity preservation, which could allow species to track changing habitat conditions. In this context, to evaluate resilience of a metacommunity is really important to consider the degree of connectivity of the network.

We agree with the Reviewer on the importance of considering the degree of connectivity among the networks. We have included a new analysis based on graph theory to characterize the spatial structure and connectivity of metacommunity networks (see *Metacommunity networks and connectivity*). Following previous work (refs 53,54), we derived a percolation graph from each metacommunity and quantified two centrality metrics on these graphs – degree and closeness centrality – as proxies for direct and global connectivity, respectively. We derived percolation graphs from geographic (least-coast path distance by the sea) and biological (using Jaccard dissimilarity) distances for each metacommunity, using the minimum distance that maintained all sites connected as the percolation threshold. We expected that in a dispersal-limited metacommunity geographically isolated sites with low closeness centrality should also be more biologically distinct (high community dissimilarity and low degree centrality) than more central sites in a percolation graph. This should result in a significant positive relationship between closeness centrality measured on a physically-derived graph and degree centrality measured on a biologically-derived graph. In contrast, in a well-mixed metacommunity, geographic distance among sites should have no bearing on biological distance. We found no significant relationships

between centrality measures in any of the metacommunities analyzed, suggesting that geographic isolation did not preclude biological connectivity in the metacommunities analyzed.

ii. Second, authors assume that distant sites became connected over multiple generations through stepping-stone effects within a metacommunity. This scenario is a full connected network, i.e., all communities are connected among them. However, the metacommunity framework has sense at intermediate connectivity values. Since at low connectivity values, communities are isolated, and at high connectivity value it is a big community.

In addition to the analysis described above, we also compared the average compositional dissimilarity between MPA and OA sites in each metacommunity, using the Jaccard index. Values ranged between 0.21-0.74 and 0.26-0.64 in MPAs and OAs, respectively, suggesting that fish dispersal neither completely differentiated nor completely homogenized metacommunities. Although evaluating the ‘intermediate connectivity hypothesis’ framed by the Reviewer remains a difficult task, we feel that this analysis and the previous one provide robust evidence that sites were biologically connected and thus formed metacommunities.

iii. Third, in figure 1 the connectivity among sites is mentioned, but it is not clear how this is considered in the analysis. It is associated to the size (number of communities) and the maximum distance between sites. Why not a SEM for the metacommunity scale? Each metacommunity is an ecoregion? In the discussion (that begins in Line 335) something is slightly mentioned about the location of sites, and the flow individuals through network connectivity. However, is not herein explicitly considered.

The new analyses described above should now clarify the meaning of connectivity in Fig. 1. We also clarified that metacommunities are defined by the sites in each ecoregion (lines 241-242). We have considered running SEMs separately for each metacommunity, but we believe our mixed-model SEM is more powerful and provides more cohesive results that would otherwise become too fragmented if presented in separate analyses.

Thus, I think it would be great to incorporate analysis that really evaluate the metacommunity scale and improve in the discussion the metacommunity framework.

We agree and believe our new analyses fully address these points, which we now discuss in more detail (lines 398-412).

2. Environmental heterogeneity refers to temporal fluctuations of the metrics used? It could be confused with the variability of an environmental filter in the time, but I think is not the case. It is a bit confuse since is mentioned in the introduction and never more.

Environmental heterogeneity refers to changes in abiotic conditions, temporal fluctuations in environmental drivers that can affect stability and asynchrony at different levels of biological organization. We have replaced “heterogeneity” with “variability”, to better reflect changes in the abiotic environment (line 142).

3. As it is showed in Figure 3a and 3b, I found interesting that similar direct positive effects were observed in MPAs and OAs (except for the connection between remoteness and alpha stability). Functional richness is positively associated with species stability in protected and non-protected areas. But what promotes functional richness? That is, basic mechanisms that drive ecological relationships independently if it is the MPAs or OAs. The main differences are in the negative effects of MHWs in OAs. So how the results support the hypothesis that MPAs promote reef fish stability at the community level, different from the mechanisms in OAs? And also, why not considered one analysis incorporating the identity of the area (MPAs or OAs). What is the meaning of that well-enforced MPAs that authors said that buffer the impact of MHWs on species and community stability?

Our conclusion that MPAs support stability stems from the evidence that MHWs decrease stability in OAs, but not in MPAs. The other way to communicate this result is to say that MHWs have negative effects on stability in OAs, but not in MPAs. We prefer to maintain our original statement, since it emphasizes the positive role of adding protection as a management strategy (as opposed to the *status quo* of OA). We agree there is no clear evidence of important drivers of functional richness in the analysis, and most of its variation remains unexplained. Thus, we cannot say what promotes functional richness from our results. We could have conducted just one SEM with protection level as a factor, also including other covariates, but the complexity of the analysis increases rapidly as more covariates are added, complicating the interpretation of results. We prefer to leave the analysis as it is, since it provides a solid, clear outcome which can be readily communicated.

4. Remoteness is a proxy of human pressure, measured herein as the travel time to large cities. Which was the range of remoteness used? Was it similar in both areas? In addition, the relationship between remoteness and functional diversity in the MPAs was evaluated? Less functional richness could be expected in more isolated communities.

Travel time to large cities was very similar for MPAs and OAs (mean: 0.504 for MPAs and 0.530 for OAs; range: 0.006-7.54 for MPAs and 0.002-9.70 for OAs). The link from remoteness to functional richness was originally included in the analysis for MPAs, but this resulted in a significant ($p < 0.05$) Fisher's *C* score, indicating that the model was not properly specified (see Methods: lines 667-670). Since the link was not significant, it was subsequently removed and this improved the model fit, making Fisher's *C* score not significant ($p > 0.05$).

Minor comments

1. Authors propose several hypotheses about the drivers of resilience in MPAs and OAs in the introduction at local and metacommunity scales. Maybe a table or a conceptual diagram could be incorporated to clarify and synthesize the main hypothesis.

Agree. We have included a table to summarize the main hypotheses evaluated in the study.

2. Positive and negative effects in the SEM have the same colors that MPAs and OAs.

We think this does not impair interpretation, rather, we believe blue links reinforce the message that protection has positive effects (the opposite for orange links).

3. *Some details of the analysis about thermal sensitivity trends in Result section, could be moved to methods.*

We have carefully considered this option. Due to the structure of Nat. Comm. articles, with the Methods section at the end, we think it might be better to leave at least some details in the Results section, as most readers will probably go through that before looking at methods.

4. *Functional richness was defined by three axis of the PCoA, how they were used in the SEM?*

The PCoAs define a multidimensional trait-space. We reduced it to a three-dimensional space and then quantified functional richness as the proportion of such space occupied by all species in a site. This generated one estimate of functional richness per site, which we used as a covariate in SEMs. We now better clarify this procedure at line 346.

5. *How open areas are defined? They said that are areas subjected to some form of extractive use.*

Open areas are sites subjected to fishing and other human extractive uses. This has been clarified (lines 108-109).

REVIEWER 2.

It is important to clarify whether Marine Protected Areas (MPAs) is effective to protect biodiversity. Benedetti-Cecchi et al. compiled a large timeseries of population abundance for reef fish species from both protected and open sites world to assess the effects of MPAs to increase stability from populations to metacommunity, particularly in the response to the marine heatwaves. This study shows support that MPAs can increase temporal abundance stability of populations, communities and metacommunities. Overall, I think this study can fill an important gap in assessing the effectiveness of protected areas. However, some parts of the manuscript, particularly the method section, need to be largely clarified and improved.

We thank the reviewer for the constructive comments and suggestions to improve the manuscript.

General comments:

1. *Marine heatwaves (MHWs) is an important term/variable to fully understand this manuscript. I think it should be clearly defined in the main text. In the method section, it was still not clear how marine heatwave was calculated. It was stated that the climatology was derived from the 30-yr period 1982 to 2012, but the population timeseries were sampled from 1992 to 2021. Does it mean that it is not necessary the time periods of population trends and marine heatwaves need to be matched in the analyses? The calculation of marine heatwaves needs a threshold, which was stated based on “seasonally varying 90th percentile” daily sea surface temperature (SST). It needs more details, such as how seasons were determined, how the climate time series was used in calculating the thread, etc.*

We thank the reviewer for giving us the opportunity to clarify the definition and calculation of MHWs. We have used a standardized approach, as implemented in pivotal studies on the topic (see refs. 36-37), where MHWs are defined as SST anomalies that exceed a seasonally varying

climatological threshold, for at least 5 consecutive days. The climatological threshold is obtained from the 90th percentile of SST variation calculated over a recommended climatological period of at least 30 years. SST data were available from 1982 and thus the period 1982-2012 covers the recommended 30-year time span for calculating the climatology. Thus, for each focal calendar day from 2013 onwards, there are 30 data points of previous SST values from which a climatological mean and threshold specific for that focal day can be calculated. The mean and the threshold are calculated within a 11-day window centered on the focal calendar day. The mean and threshold are further smoothed by applying a 31-day moving average to filter out short-term temporal variation. A MHW is detected when SST values are above the threshold for at least 5 consecutive days. The key point of this procedure is that the threshold is not fixed, but changes from day to day. It is defined as a *seasonally*-varying threshold to emphasize that seasonal fluctuations do not affect the identification of MHWs, since they are identified by comparing the SST for each calendar day to a distribution of past SST values for that day. It could have been termed *day*- or *time*-varying threshold, but the language emphasizes seasonal cycles as the major source of confusion had MHWs been defined with respect to a fixed threshold that does not change during the year.

We have clarified the approach in the main text, as recommended by the reviewer (lines 125-127) and better articulated the matching of MHWs with timeseries of fish abundance in the Methods (lines 484-490).

The intensity of marine heatwave was calculated as the difference between the observed SST and the threshold SST. If a given year had multiple marine heatwaves, how the intensity was calculated? If a mean difference between the observed SST and the threshold SST was used and two sites had the same difference, how to distinguish two sites with one and two marine heatwaves events?

This is another important point that we have clarified. We agree with the reviewer that a site that received two MHWs may not be distinguished from a site that received one MHW using mean intensity. In this respect, cumulative intensity may provide a better measure of MHWs. We repeated the analyses using cumulative intensity and results did not change. We have mentioned this sensitivity analysis in the text (p. lines XX) and included the outcome in Supplementary Information (Supplementary Fig. 4). Nevertheless, we retained mean intensity as the main descriptor of MHWs, since SEMs based on cumulative intensity of MHWs, although providing qualitative similar results of those using mean intensity, yielded a significant Fisher's *C* score, indicating that models were not properly specified. SEMs including mean intensity to characterize MHWs were properly specified (Fisher's $C > 0.05$). Thus, mean intensity of MHWs has been used throughout for consistency.

2. Figure 4 shows the relationship between abundance MHW intensity for different groups of species. The figure legend describes it shows "temporal trajectories". Did the "abundance" indicate the mean abundance of a group of species across years or the temporal trends of abundance over years? If it indicated absolute abundance (rather than abundance trends), I think the spatial variation in abundance should be determined by many environmental variables, which were not considered presently. Also, I am sure whether the abundance is comparable when different methodologies were used in collected samples.

The figure shows changes of mean fish abundance in relation to MHWs, not temporal trajectories. We have corrected this in the figure legend, and we thank the reviewer for spotting this mistake. We agree with the reviewer that many environmental variables can affect fish abundance, but this aspect has no influence on our analysis and conclusions. The main purpose of our study is to test whether MPAs could support higher fish abundances and promote stability under warming conditions by allowing thermally resistant species to attain large population sizes compared to thermally sensitive species. We believe MHW intensity is the relevant covariate to test this specific hypothesis and although other covariates could have been included, we prefer to maintain the analysis as simple as possible.

The analysis on fish abundance includes an offset to control for the different sampling efforts among datasets used, as the analysis on alpha stability (see our reply to the next question).

3. Sampling effort was largely varied across sites in the compiled dataset, as the authors stated. The sampling effort is expected to affect most biodiversity measures. To control for sampling effort, an offset was used in all models. I don't think it was an effective way for models with stability and asynchrony as response variables. In the manuscript, stability was measured as the inverse of coefficient of variation (mean/sd). If the abundance collected is proportional to sampling areas (efforts) across years, both mean and sd will be proportional to sampling areas but the coefficient of variation will not be affected. Asynchrony was measured using correlation between samples (population/species/community) and it will also not be affected by sampling areas if the abundance collected is proportional to sampling areas. I don't say stability and asynchrony are independent of sampling effort because larger sampling areas are expected to have smaller variation over time when heterogenous parts of a large sampling area can compensate in the fluctuation. However, the sampling efforts cannot be addressed by directly including the sampling effort as an offset. I think the sampling efforts can be standardized for each population/species/community first and then stability and asynchrony can be calculated based on the standardized data.

We agree with the reviewer that mean fish abundance and standard deviations are proportional to sampled area. However, nothing guarantees this relationship, hence stability, follow the same patterns in MPAs and OAs and that these patterns remain constant across levels of MHW intensity. Indeed, our results show that this is not the case (Supplementary Fig. 8). The approach proposed by the Reviewer (i.e., standardize fish abundance by sampling effort first and then calculate stability/asynchrony), would be not effective since sample area (our metric to standardize sampling effort) is a constant in each site and dividing fish abundance by a constant would make no difference: we would obtain exactly the same values of stability and asynchrony as those obtained by analyzing the original data. This is a well-known property of the coefficient of variation that also applies to its inverse, our measure of stability (ref. 95). The same applies to asynchrony, since dividing timeseries of fish abundances by a constant leaves the relative differences among timeseries unchanged. We believe standardizing stability and asynchrony directly in statistical models, as we have done, effectively accounts for what the reviewer is concerned about. This approach is appropriate and common among studies that, like ours, combine data from multiple programs with varying sampling effort, such as in bird surveys (e.g., Wauchope, H. S. *et al.* Protected areas have a mixed impact on waterbirds, but management helps. *Nature* **605**, 103-107 – 2022). Thus, we have retained our original analysis including the offset in models, but we have provided a full justification of our approach adding a new section to the Methods (*Controlling for*

sampling effort, lines 681-720) to explain what an offset is and how it is used to scale the response variable. We have also highlighted the scaling effect of the offset in the main text (lines 153-156).

In short, an offset is a fixed quantity associated with each observation that is used to scale the response variable, such that its influence is accounted for in the model. The offset is added to the linear predictor with a fixed coefficient of 1 (i.e. no regression coefficient is estimated for an offset) and the scaling is simply achieved by subtracting the offset from the response variable. When both the response and the offset variables are log-transformed, the analysis is a model of log-response ratio – e.g., a rate of change in stability or asynchrony by unit of sampled area. With standardization (scaling and centering) after log-transformation, the offset is subtracted from the response variable. We have shown that the two transformations produce essentially the same results (this is shown for stability in the newly added Supplementary Fig. 5), but we have used standardized response and offset variables in the analyses, since standardization improved model fit and data visualization.

4. The manuscript assessed how the effect of MPAs on stability and asynchrony varies related to spatial scale. The authors calculated the distance between pairwise sites in MPAs and used the maximum distance as the spatial scale. If multiple sites are not randomly distributed in MPAs, e.g. five sites distributed closely but one site distributed from the other five sites, the maximum distance can't well represent the distance (spatial scale) among these sites. Also, I don't fully understand this sentence "For each MPA site we identified all other sites within the defined spatial scale and performed all possible pairwise comparisons separately for each level of protection". Because the spatial scale was defined as the maximum distance between MPA sites, all pairwise MPA sites will be within the defined spatial scale. FOR OAs sites, not all sites will be within the defined distance from each MPA site. In the described method, only OAs that were not far from the MPA sites were selected. However, the distances among OAs should be smaller and may be not comparable with the distance among MPA sites.

This is another aspect that required clarification and we thank the Reviewer for pointing this out. The comment is correct, when considering the maximum distance between MPAs, all protected sites will be within that distance by definition. We have rephrased the Method section to clarify this point (lines 750-756). However, to address the Reviewer's comment, we have performed a sensitivity analysis by matching MPA and OA sites within a spatial scale of 50-100 km. This range of distances is intermediate between the maximum distances separating MPAs in metacommunities, with 100 km representing a potential upper limit of direct fish dispersal (see refs 40,55). The analysis was repeated for the six metacommunities that had sites in the 50-100 km distance range and results are essentially the same of those from the maximum distance analysis. This has been reported in the main text (lines 321-330) and in Supplementary Fig. 12. We hope this sensitivity analysis and the visualization of metacommunity networks (see response to the first comment of Reviewer 1 and Supplementary Figs. 9 and 10), have clarified the point.

Specific comments:

Line 46: Fig. 2 showed a positive relationship between stability and remoteness. A higher remoteness means lower human pressure. so, the statement "We find that the intensity of human pressures is positively related to local stability only in protected areas" is incorrect.

Rephrased as: “We find that local stability is positively related to distance from high densities of humans only in protected areas”.

Lines 98-102: Does it mean high functional richness increases the asynchrony among species and further improves total abundance stability?

We now better clarify that high functional richness should increase the range of species responses to environmental fluctuations, which should resonate on asynchrony and then stability. This is presented here as a testable hypothesis, not as a conclusive statement.

lines 131-135: I can't understand how “Thus, by reducing direct human disturbances, MPAs should mitigate population and species fluctuations, increasing gamma stability” can be expected.

We now better clarify that by removing fishing and other direct human disturbances, MPAs should provide less variable environments compared to OAs, in which case also populations and then total fish abundance should fluctuate less, resulting in greater overall stability. Again, this is proposed here as a testable hypothesis not as a conclusive statement.

Line 178: The subtitle is not easy to understand what will talk about.

We couldn't come up with a more informative subtitle, sorry.

Lines 184-186: How to interpret fish abundance increases with MHW in protected areas?

We have indicated in the text that MHWs are more intense at higher latitudes and many species are expanding their ranges towards the poles in response to global warming, with MPAs likely providing safe places to allow these species, especially carnivores, to establish large populations (line 369 and lines 376-385).

Line 217: “In this section” is not necessary here in the results part.

Removed.

Lines 254-256: Why effect sizes not deviate significantly from zero can indicate higher gamma stability in OAs than in MPAs?

We have added “the latter ..” before “indicating” to reflect that only negative effect sizes indicate higher gamma stability in OAs than MPAs (line 312).

Line 276: It is not clear that “stronger asynchronous fluctuations” in MPAs.

We have replaced the term “promote” with “maintain” to tone down the statement. However, the results provide solid and consistent support to the conclusion that MPAs buffer the impact of MHWs on asynchrony compared to OAs. For instance, Fig. 2h show a significant negative relationship between asynchrony and MHW intensity in OAs, whereas in MPAs the trend is

negative, but not significant. Fig. 3 shows a direct and significant negative path from MHWs to asynchrony in OAs, but not in MPAs (Fig 3b vs. Fig. 3a).

Lines 288-289: The statement “fishes that experienced warming events beyond their upper realized thermal limit (STI below threshold) increased in abundance maximum intensity of MHWs” was not strongly supported by results (Fig. 4)

We now make explicit reference to the lower panels of Fig. 4. These show that fish abundance is larger in MPAs than OAs over a range of MHW intensities, supporting our statement.

Lines 308-313: I don't find these sentences closely connected to the topic discussed in this manuscript.

We prefer to leave this section as it is because we believe it improves the clarity of the MS by providing a mechanistic explanation why, in contrast to grazers, only fish carnivores with a STI above threshold (thermally-resistant species) established a positive relation with MHWs. Large fishes have lower thermal tolerance than smaller ones and carnivores are, on average, larger than grazers.

Lines 443-444: How many species have no species-level trait data?

8% of the taxa could not be resolved at the species level; this has been now clarified (line 536)

Lines 490-491: This equation is not absolutely correct. Possible corrections are to write out the $SP[stab, i]$, or remove $sum[j]$.

We thank the Reviewer for having identified this mistake. We have removed $sum[j]$ from the equation.

Line 521: This equation should further multiply $u[i]/u$ in the brace.

Done, thank you.

Line 555: A typo, “and” rather than “ad”.

Corrected.

Line 568: “the model for species asynchrony included species stability”: in all results reported, there was no relation between species stability asynchrony. Did the relationship was removed in the model selection?

We did not model this relationship since we had no *a priori* hypothesis about the direction of a causal path between these variables (lines 659-661) and their covariance was not significant.

Line 594: What is the cost layer/raster/data to calculate the least-cost distances?

We now better clarify that this is provided by land masses

Line 654: It is necessary to provide more details about the null model to clarify how the model works. Currently, only “cyclic shift algorithm” was given. Is the abundance of a given species in a given site cyclically shifted or does the total abundance in a given site shift?

We applied the cyclic shift algorithm to each species independently in each site. We have clarified this in the text (lines 824-825).

Figure 2: Why R² was present in panels b, f, j, k, but not in other panels?

Supplementary Fig. 2: Why R² was present in panels a, f, i, l, but not in other panels?

Supplementary Fig. 3: Why R² was present in panels a, b, e, f, but not in other panels?

In these figures, panels illustrating responses for the same variable have the same R² since they originate from the same model. We have reported the R² only in the first panel for each response variable for clarity. This is indicated in the legend of the figures: (R², indicated only in the first panel for each response variable).

REVIEWER 3

This study aggregated fish UVC data from 12 survey programs around the world with the objective of evaluating the stability of assemblages in the face of stressors like warming and human impacts and the degree to which marine reserves can mitigate this. The overall conclusion was that assemblages did seem to be more stable in marine reserves than in areas open to fishing, even mitigating effects of warming and being positively related to fish abundance in reserves. In general I really like the approach taken here, looking at system stability. I think these results would have huge benefits for our understanding of the utility of marine reserves. However I outline a number of issues below, largely to do with unknowns about the data aggregation and its challenges, that could have large impacts on these overall conclusions.

We thank the reviewer for the appreciation of our work and for providing constructive comments to the manuscript.

Major comments

I was concerned about how area assessed in each of the various different survey programs was handled when data was aggregated. Certainly these different programs have different areas surveyed, sometimes greatly. And programs like RLS and the AIMS LTMP have different methods to capture different suites of species within their program which have very different survey belt widths. So while I can see the logic in adding areas surveyed for each replicate as a co-variate, it did seem more complicated than needed. Why did you not just convert all values to density and then proceed from there? There is of course an often unrecognized assumption when doing this, namely that abundance scales linearly with area surveyed, but that seems to be a very safe assumption and thus this approach is widely used. So in theory including area as a co-variate should work but here all response variables are standardized first, and area is log transformed prior to this. This would have the effect of decoupling this relationship and I'm guessing you would not get the same answer as had one just used density given the log transform would affect some

area values more than others. As this is the basis for everything else that's done in this study, it's a pretty fundamental issue to either clarify or to fix.

We assessed the total area sampled in each site by multiplying the area of a transect by the number of transects surveyed in a year (including repeated sampling dates, when present). The approach suggested by the Reviewer to divide abundance by sampled area would not be effective. We report below our response to a similar comment (comment #3) made by Reviewer 2. Sample area (our metric to standardize sampling effort) is a constant in each site and dividing fish abundance by a constant would make no difference: we would obtain exactly the same values of stability and asynchrony as those obtained by analyzing the original data. This is a well-known property of the coefficient of variation that also applies to its inverse, our measure of stability (ref. 95). The same applies to asynchrony, since dividing timeseries of fish abundances by a constant leaves the relative differences among timeseries unchanged. We believe standardizing stability and asynchrony directly in statistical models, as we have done, effectively accounts for what the reviewer is concerned about. This approach is appropriate and common among studies that, like ours, combine data from multiple programs with varying sampling effort, such as in bird surveys (e.g., Wauchop, H. S. *et al.* Protected areas have a mixed impact on waterbirds, but management helps. *Nature* **605**, 103-107 – 2022). Thus, we have retained our original analysis including the offset in models, but we have provided a full justification of our approach adding a new section to the Methods (*Controlling for sampling effort*, lines 681-720) to explain what an offset is and how it is used to scale the response variable. We have also highlighted the scaling effect of the offset in the main text (lines 153-156).

In short, an offset is a fixed quantity associated with each observation that is used to scale the response variable, such that its influence is accounted for in the model. The offset is added to the linear predictor with a fixed coefficient of 1 (i.e. no regression coefficient is estimated for an offset) and the scaling is simply achieved by subtracting the offset from the response variable. When both the response and the offset variables are log-transformed, the analysis is a model of log-response ratio – e.g., a rate of change in stability or asynchrony by unit of sampled area. With standardization (scaling and centering) after log-transformation, the offset is subtracted from the response variable. We have shown that the two transformations produce essentially the same results (this is shown for stability in the newly added Supplementary Fig. 5), but we have used standardized response and offset variables in the analyses, since standardization improved model fit and data visualization.

Another issue with data aggregation that is not addressed but would seem to be of fundamental importance is the different levels of taxonomic specificity and more importantly, scope, of the different surveys. I'm not familiar with all these survey programs but I do know, for instance, that RLS works to capture cryptic species, while AIMS LTMP does not. However, LTMP does use two different belt widths to capture more versus less mobile taxa very specifically. I'm not familiar with ReefCheck methods but in looking over the information at the link provided in the manuscript, the data captured is mainly at the family level but doesn't seem to include important herbivores like Acanthurids. While I appreciate the focus of the study was not on diversity or richness, these differences in taxonomic coverage would seem likely to influence the potential for stability or asynchrony as I'd imagine both are less likely the more taxa one considers. And these differences

in taxonomic coverage would definitely affect the analysis based on total numbers, such as the trophic guild analysis and anything to do with functional richness.

We thank the Reviewer for raising this point. We have added a new analysis using sample coverage to evaluate whether fish communities were adequately sampled by the different monitoring programs with varying sampling efforts and size of sampling units (transects, cylindrical plots). In particular, we have evaluated whether there was any systematic difference between MPAs and OAs, since this was the main focus of our analysis. We have added conceptual clarifications in the main text (lines 173-177) and in the Methods (lines 801-812): sample coverage is a measure of sample completeness and gives the proportion of the total number of individuals in the community that belong to the species represented in a sample of that community (ref. 48). We compared sample coverage between MPAs and OAs for different size categories of sampling units (transects or cylindrical plots) and we found that fish communities were sampled with comparable accuracy in MPAs and OAs (Supplementary Fig. 6). Only transects in the size category of 180 m² indicated larger completeness in OAs than in MPAs. These sampling represented a small fraction (2.2%) of the total samples and removing them from the analysis did not change the results. Since systematic differences between levels of protection were negligible, and all sampling programs included both, we believe differences in taxonomic coverage did not affect our comparisons of stability and asynchrony between MPAs and OAs.

The previous two points, in combination with the unexpected result that thermally sensitive species were more common in areas where MHWs were more intense (which tend to be away from the tropics...a point made by the authors) raises concerns as to how much of the patterns observed in the aggregated data set are just the result of regional differences due to different survey methods (different areas and taxonomic scope). This isn't discussed at all. I see that study ID was included but nowhere are the results of this, or the area coefficient, reported. Similarly, is either stability or asynchrony influenced by the number of taxa (time series) considered? If so, would you expect any latitudinal trends, with fewer species as you move away from the tropics? Can this explain any variation in the data.

Actually, controlling for sampling effort in statistical models and ensuring comparable sampling completeness between MPAs and OAs make our results robust and independent from differences in sampling methods. We emphasize that our models also included a random intercept for study ID to further control for possible generic differences in methodology among monitoring programs. Results from alternative analyses are reported in full in Supplementary Tabs 1-3. The variance component among sampling programs is indicated by the parameter τ_{00} . We have now clarified in the legend of the tables that this parameter is the variance associated with the study ID intercept. The offset (sampled area) has no coefficient. As articulated above, the offset is added to the linear predictor with a fixed coefficient of 1.

The minimum for five years in a data series does seem short though I appreciate there is probably no perfect answer here. Though I was surprised to see the justification for this time period is based on studies in grassland assemblages. Is the scale of temporal dynamics the same as with fish assemblages? That would seem important to clarify in order to justify basing this time period of such a seemingly different system.

We have modified and expanded the relevant section in Methods to better clarify this point (lines 452-454). Unfortunately, there is no objective criterion to determine the length of timeseries to analyze population trends, stability and asynchrony. Two previous studies (refs. 28, 50) used five years as the minimum requirement to analyze stability in grasslands and in a range of terrestrial animals (arthropods, birds and bats). The five-year criterion has also been adopted in a global analysis of population trends in vertebrates, including marine fishes (Daskalova et al. 2020. Nat. Comm. 11:4394). Thus, a consensus around this rule of thumb seems to emerge in the literature for a wide range of taxa. In the absence of a clear rationale, we have adopted the same criterion here.

I found the acronyms quite confusing as there was no consistent system to them. That said, I did spend 10min trying to come up with a better system and am not confident I did anything more useful. So it's not easily solved but will definitely be an obstacle to the uptake of information here. One observation I'd make is I'm no sure using alpha and gamma helps clarify things. I appreciate those come from diversity standards, but really you could refer to 'site stability' rather than alpha and then 'species stability' makes more sense and both are seen as cutting different ways across the data at the local scale. Then you can have 'average site stability'. Similarly for gamma it could just be meta-community (or MC) stability.

We appreciate the help of the Reviewer to simplify the language. The large number of acronyms stems from the variety of metrics used to quantify stability and asynchrony for local communities and metacommunities, all of which have their meaning and interpretation. This makes the system inherently complex. Nevertheless, we prefer to maintain the standard terminology used in the space of stability analyses. This terminology is well established in the literature (e.g., refs. 25-28, 32, 88) and we have defined our terms in the first paragraphs of the manuscript (line 90 for alpha stability and line 98 for gamma stability).

Minor Comments

Ln 551: can you be more clear over what data set the z-scores were generated. I assume the whole data set for each variable, but good to be clear (in case, for instance, it was by regions...which would seem unlikely).

We now better clarify that standardization (scaling and centering) was done for each response variable across the whole dataset. For example, alpha stability was scaled and centered over the 1104 estimates (one for each site), before analysis (line 644).

Figure 1. I love the approach of this figure, and its definitely needed. But I still found it quite confusing after a number of reads. I believe this is because what's shown are time series but really the measures discussed are just derived from the variance and means of these time series in different ways. I wonder if you could restructure this such that you start with time series as in, say, panel c. You then represent the temporal mean and SD on those and then branch off those values in different ways to show how all the other measures are calculated?

We recognize this is a dense figure with a lot of information pertaining to components and organizational levels of stability and asynchrony. We have slightly modified the figure by

swapping ASA with SSA between panels c and d for consistency and provided an explanation for the pink and green ovals (sites) in the legend. We have also rephrased the description of SSA to match the definition provided in the Methods. However, we could not follow the Reviewer's suggestion of deriving all measurements from a single panel, since the various stability and asynchrony components combine timeseries in different ways (e.g., abundance of individual species in a site, total population abundance averaged across species, total community abundance, etc.) and we could not figure out how to relate all these components to a single timeseries. Nevertheless, our explanation of the site symbols clarifies how the timeseries were aggregated and how they related to the various stability and asynchrony measures, hopefully making the figure more accessible. We would be happy to consider further suggestions to improve the figure, but we feel the reader will need to spend some time with it, no matter how we structure it.

REVIEWER COMMENTS

Reviewer #1 (Remarks to the Author):

Comments to Authors.

I have read the authors responses and the revision of their manuscript, and I think that, the authors have adequately addressed most of my concerns. On the whole, it is much improved. However, I still have some comments about the new analysis based on graph theory.

Methods and Supplementary Figure 9 and Figure 10. I am afraid that the graph is not a percolation graph; it seems to be a minimum spanning tree (MST). The MST connects every community with the shortest path length—i.e., with the minimum number of links to ensure that all patches are connected to a single graph, and without the occurrence of loops. The percolation network is defined as a graph in which patches are connected when the distance between them is less than a threshold distance, which is the minimum distance (i.e., the percolation point) at which all patches are connected in a single network. All the results about connectivity metrics and interpretation depends on that the graph estimation will be ok.

I didn't understand this sentence: "Although connectivity is rarely assessed in studies of gamma stability..." Why? Maybe authors could incorporate a reference.

Please, added some reference to the centrality metrics, lines 305-310.

How were the weights considered to estimate degree centrality? Because they are distances, so the weights should be $1/\text{distance}$. In case of closeness the metric estimation in the igraph package in R considers weights as distances, so it will be ok as it is.

In this case, low closeness centrality means geographically isolated sites, while low degree centrality means biologically distinct. Different metrics of centrality emphasize different spatial scales, and depending on the graph structure the metrics could be or not correlated. What I don't understand is why compare metrics that have different weights. In stead of degree centrality, why not use a beta diversity metric.

Finally, herein I add some references that could help for graph tools (specifically centrality metrics and graph estimation):

Urban, D. and Keitt, T. H. 2001. Landscape connectivity: a graph-theoretic perspective. *Ecology* 82: 1205-1218.

Economu, E. P. and Keitt, T. H. 2010. Network isolation and local diversity in neutral metacommunities. *Oikos* 119: 1355-1363

Rozenfeld, A. F. et al. 2008. Network analysis identifies weak and strong links in a metapopulation system. *Proceedings of the National Academy of Sciences (USA)* 105: 18824-18829.

Borthagaray, A. I. et al. 2015. Effects of metacommunity network on local communities structure: from theoretical predictions to empirical evaluations. - In: A. Belgrano et al. (eds), *Aquatic Functional Biodiversity*. Elsevier.

Reviewer #2 (Remarks to the Author):

The authors did a good job in revising the manuscript and have addressed most of my concerns (reviewer 2). However, I was not convinced about the approach to controlling for sampling efforts.

An offset of the sampling area was added as a predictor in model fitting. As stated in the manuscript, the offset means a fixed coefficient of 1 for area. So, this approach assumes that the stability and asynchrony proportionally increase with the sampling area. I don't think the assumption is rational. I agree that when both the response and the offset variables are log-transformed, the scaled response variable ($\log(\text{stability}) - \log(\text{area})$) can be expressed as a log-

response ratio and interpreted as stability per unit of area ($\log(\text{stability}/\text{area})$). However, is it meaningful to standardize stability by dividing the area? Some variables, such as the number of individuals, can be standardized by area because it proportionally increases with area. For stability, however, it seems not reasonable to standardize by dividing the area.

Regarding standardizing abundance data to density, I agree that the sampled area was a constant at any given site and that the stability based on the density data within a given site would result in the same value as that based on the original data. However, for stability and asynchrony variables that need to be averaged across multiple sites, the results based on the density data may be different from those based on the original abundance data, because site weight changes when density data is used.

A few minor comments:

Lines 135-141: It hypothesized that environmental heterogeneity would increase temporal fluctuations, while environmental variability increases asynchrony. Because environmental heterogeneity usually means spatial heterogeneity, it may affect spatial asynchrony; while environmental variability in this study means temporal variability, it may affect temporal fluctuations.

Line 143: "Table 1" rather than "Tab1. 1".

Table 1: In "MPAs are more stable than OAs": "than" rather than "that".

In "Greater stability (lower fluctuations) in abundance of individual species and greater functional richness increases stability in MPAs compared to OAs": "increases" is redundant.

Figure 2: R2 should be indicated only in the first panel for each response variable, but it was missing for species asynchrony (panel h) and redundant in panel k.

Lines 1166-1167: "Positive (negative) effect sizes indicate larger (lower) stability or asynchrony in MPAs than OAs": Both effect sizes of MPAs and OAs were shown in panel c, so what did the sentence mean?

Line 188 in the supplementary information: line numbers were overlapped with table content.

Reviewer #3 (Remarks to the Author):

I thank the author for their thorough attention to all comments provided, including mine. Overall I'm satisfied that my concerns have been addressed. I do have a few remaining comments.

Thank you for the thorough explanation of the technique used to adjust for the sample effort. I would note that in your explanation, line 686 in the tracked changes version, you indicate using density was not a viable option because sampled area was a constant at any given site. I don't question that the offset achieves the desired outcome but I am confused by this explanation. Assuming you mean all the surveys from a set of sites are of the same area so they are all divided by a constant, is that not the same issue for the offset...they all wind up with the same offset value. Isn't the issue here standardizing the surveys across areas where the area sampled was not the same? Or maybe are you saying that within the metrics you calculated, you never actually merge data series from different survey programs...even at the metacommunity level?

I'm still not 100% convinced that the fact these surveys have different target groups of spp does not affect the results. I appreciate the estimation of completeness is a robust way to assess that different survey areas are not affecting the degree to which the richness of the surveyed community is captured but does this not depend completely on the taxonomic scope of the survey in the first place. So you could get similar levels of completeness for surveys which look to assess Scarids versus Acanthurids only. Given these different surveys seem to have, in some cases, quite different target lists, they could all be doing an equally good job of achieving saturation on the

rarefaction curve, and thus good completion scores, but that may not mean you can compare metrics which assume the whole assemblage was surveyed. And as I said, the worry here is that overall richness, and by extension functional redundancy, do affect the metrics you derive on stability...at least I believe that was said. So I think it would be important to verify that the differences in overall richness or perhaps just functional richness, of the target list for these surveys is not affecting these results in any way. Of paramount importance would be the OA versus MPA comparisons but perhaps the biggest effects might be on the analysis which look at the geographical spread of metrics and the relationships to MHWs, which may also have geographic bias to them, given different areas tend to be covered exclusively by one survey type or another. It would be good to see some summary of the taxonomic scope of these different surveys and then some analysis to verify these differences are not affecting the results.

We provide a point-by-point response to the second round of comments made by the reviewers. Please, note that the original comments are in *italic*, whereas our response is in plain text. Where appropriate, we indicate the lines in the manuscript where we have made changes. **Please, note that lines refer to the position in the word document with track changes, not in the pdf with accepted modifications.**

REVIEWER 1.

I have read the authors responses and the revision of their manuscript, and I think that, the authors have adequately addressed most of my concerns. On the whole, it is much improved. However, I still have some comments about the new analysis based on graph theory.

We thank the Reviewer for this positive comment.

Methods and Supplementary Figure 9 and Figure 10. I am afraid that the graph is not a percolation graph; it seems to be a minimum spanning tree (MST). The MST connects every community with the shortest path length—i.e., with the minimum number of links to ensure that all patches are connected to a single graph, and without the occurrence of loops. The percolation network is defined as a graph in which patches are connected when the distance between them is less than a threshold distance, which is the minimum distance (i.e., the percolation point) at which all patches are connected in a single network. All the results about connectivity metrics and interpretation depends on that the graph estimation will be ok.

The Reviewer is absolutely right. We considered a percolation graph in first place, but the graph was less clear than the minimum spanning tree based on consultation across our author team, so we eventually opted for the latter. Comparisons of metrics from distance- and biologically-derived networks lead essentially to the same conclusions whether one considers percolation or MST graphs. We have replaced ‘percolation graphs’ with ‘minimum spanning tree graphs’ throughout the text.

I didn't understand this sentence: "Although connectivity is rarely assessed in studies of gamma stability..." Why? Maybe authors could incorporate a reference.

In our reading of the literature making use of the same stability framework we applied here, we noticed that communities and metacommunities have been distinguished somewhat arbitrarily without reference to any formal analysis. For example, Wilcox et al. (Ecol Lett 2017, p. 4) stated: “We treated those plots as individual communities, and collection of plots as metacommunities”. Similarly, Hautier *et al.*, (Nat Comm 2020) state (p. 3) “In our analysis we treat each 1m² plot as a ‘community’, and the replicated subplots within a site as the ‘larger scale’, sensu Wittaker”. The larger scale referred in this sentence is the metacommunity. Indeed, most studies assume that a collection of plots at larger scales form a metacommunity, without looking at connectivity and biological differentiation. We have added these references to the text as required (line 254).

Please, added some reference to the centrality metrics, lines 305-310.

We have added references for the centrality metrics as required: Economo and Keitt (2010, Oikos) and Arim *et al.* (2023, Journal of ecology) (line 258).

How were the weights considered to estimate degree centrality? Because they are distances, so the weights should be 1/distance. In case of closeness the metric estimation in the igraph package in R considers weights as distances, so it will be ok as it is.

In our first analysis we used function *degree* from package *igraph* to calculate degree centrality; this function does not allow the specification of weights. To address the comment of the Reviewer, we have repeated the analysis by calculating weighted degree centrality using function *strength* in *igraph*. This function uses the edge attribute of the input graph as weights, which are geographic or biological distances. Since function *closeness* (with the *normalize* argument set to TRUE) weights by the inverse of distance, as recognized by the Reviewer, we have specified the weights in function 'strength' as 1/distance, where distance was the edge attribute of the input graph. In this way we have obtained the same normalization for degree and closeness centrality. This is now clarified in the Methods (lines 716-719). Repeating the analysis with weighted degree centrality did not change the results: all relationships between geographic and biological distances remained not significant (Supplementary Table 6).

In this case, low closeness centrality means geographically isolated sites, while low degree centrality means biologically distinct. Different metrics of centrality emphasize different spatial scales, and depending on the graph structure the metrics could be or not correlated. What I don't understand is why compare metrics that have different weights. Instead of degree centrality, why not use a beta diversity metric.

We agree on the meaning of the metrics and we have reviewed our description and use to ensure we are accurate in their scope/application. We posited that in a dispersal limited metacommunity representing geographically isolated sites (low closeness centrality) would also be the most biologically distinct (low degree centrality), resulting in a positive correlation between these metrics. In contrast, the two metrics should be unrelated in a well-mixed metacommunity, which is what we found for all the metacommunities investigated (see also our response to the previous comment). We have thus confirmed our analysis and interpretation is robust, addressing the original and intriguing question of the Reviewer about connectivity in metacommunities.

Finally, herein I add some references that could help for graph tools (specifically centrality metrics and graph estimation):
Urban, D. and Keitt, T. H. 2001. Landscape connectivity: a graph-theoretic perspective. *Ecology* 82: 1205-1218.
Economu, E. P. and Keitt, T. H. 2010. Network isolation and local diversity in neutral metacommunities. *Oikos* 119: 1355-1363
Rozenfeld, A. F. et al. 2008. Network analysis identifies weak and strong links in a metapopulation system. *Proceedings of the National Academy of Sciences (USA)* 105: 18824-18829.
Borthagaray, A. I. et al. 2015. Effects of metacommunity network on local communities structure: from theoretical predictions to empirical evaluations. - In: A. Belgrano et al. (eds), *Aquatic Functional Biodiversity*. Elsevier.

We made an effort to refer to Economu and Keitt in our first revision, and we now quote the paper by Urban and Keitt as it makes explicit reference to the minimum spanning tree. We appreciate this suggestion.

REVIEWER 2.

The authors did a good job in revising the manuscript and have addressed most of my concerns (reviewer 2). However, I was not convinced about the approach to controlling for sampling efforts.

We thank the Reviewer for the positive comment.

An offset of the sampling area was added as a predictor in model fitting. As stated in the manuscript, the offset means a fixed coefficient of 1 for area. So, this approach assumes that the stability and asynchrony

proportionally increase with the sampling area. I don't think the assumption is rational. I agree that when both the response and the offset variables are log-transformed, the scaled response variable ($\log(\text{stability}) - \log(\text{area})$) can be expressed as a log-response ratio and interpreted as stability per unit of area ($\log(\text{stability}/\text{area})$). However, is it meaningful to standardize stability by dividing the area? Some variables, such as the number of individuals, can be standardized by area because it proportionally increases with area. For stability, however, it seems not reasonable to standardize by dividing the area.

We have carefully reflected on the questions posed by Reviewer #2. We believe our approach provides a robust solution to standardize stability and asynchrony measures by sampling effort. To ensure this is clear, we have expanded and clarified the use of an offset and the alternative of analyzing log-response ratios (each response variable divided directly by sampled area) without the offset. In contrast to generalized models, the two approaches are equivalent for linear models (Crawley, MJ. 2013. The R Book Second Edition, p. 415 and p. 566). We show this equivalence by expanding the analysis of log-response ratios to all the variables examined in the community-level analysis (Supplementary Fig. 5). The results are very similar to those of Fig. 2 in the main text and in Supplementary Fig. 1. We note that the analysis with log-response ratios does not assign any fixed coefficient to sampling effort (sampled area), since this becomes part of the response variable. Thus, no specific assumption is needed about the relationship between stability or asynchrony with sampling effort when using log-response ratios. Nevertheless, we can show that a positive relationship exists (see the figure below), although this is not a requirement. We still opt for using the results of models including the offset since these provide a better visualization compared to log-response ratios (compare Fig. 2b-k and Supplementary Fig. 1 with Supplementary Fig. 5). We have articulated these aspects of the analysis in the main text (lines 169-172) and in the Methods (lines 695-706).

Regarding standardizing abundance data to density, I agree that the sampled area was a constant at any given site and that the stability based on the density data within a given site would result in the same value as that based on the original data. However, for stability and asynchrony variables that need to be averaged across multiple sites, the results based on the density data may be different from those based on the original abundance data, because site weight changes when density data is used.

Please, note that the offset is used only in the community stability analysis, where stability and asynchrony measures are calculated at the scale of the site and never averaged across sites. Averaging is required in the metacommunity analysis, which does not require standardization since data within ecoregions generally originate from the same sampling programs. This is now clarified in the Methods (lines 766-768).

A few minor comments:

Lines 135-141: It hypothesized that environmental heterogeneity would increase temporal fluctuations, while environmental variability increases asynchrony. Because environmental heterogeneity usually means spatial heterogeneity, it may affect spatial asynchrony; while environmental variability in this study means temporal variability, it may affect temporal fluctuations.

We have included reference to spatial heterogeneity (line 137).

Line 143: “Table 1” rather than “Tabl. 1”.

Done.

Table 1: In “MPAs are more stable than OAs”: “than” rather than “that”.

Done

In “Greater stability (lower fluctuations) in abundance of individual species and greater functional richness increases stability in MPAs compared to OAs”: “increases” is redundant.

We have changed “increases” to “increase” (plural), but left the word in place, otherwise the sentence makes no sense.

Figure 2: R2 should be indicated only in the first panel for each response variable, but it was missing for species asynchrony (panel h) and redundant in panel k.

Done

Lines 1166-1167: “Positive (negative) effect sizes indicate larger (lower) stability or asynchrony in MPAs than OAs”: Both effect sizes of MPAs and OAs were shown in panel c, so what did the sentence mean?

The sentence synthesizes both directions of effects: ‘positive’ is associated with ‘larger’, ‘negative’ is associated with ‘lower’ (in brackets). This is a common construct to shorten an otherwise very long and repetitive sentence.

Line 188 in the supplementary information: line numbers were overlapped with table content.

Thank you. This problem should be removed if the manuscript proceeds to production.

REVIEWER 3

I thank the author for their thorough attention to all comments provided, including mine. Overall I’m satisfied that my concerns have been addressed. I do have a few remaining comments.

We thank the Reviewer for the positive comment.

Thank you for the thorough explanation of the technique used to adjust for the sample effort. I would note that in your explanation, line 686 in the tracked changes version, you indicate using density was not a viable option because sampled area was a constant at any given site. I don’t question that the offset achieves the desired outcome but I am confused by this explanation. Assuming you mean all the surveys from a set of sites are of the same area so they are all divided by a constant, is that not the same issue for the offset...they all wind up with the same offset value. Isn’t the issue here standardizing the surveys across areas where the area sampled was not the same? Or maybe are you saying that within the metrics you calculated, you never actually merge data series from different survey programs...even at the metacommunity level?

We carefully considered this set of questions. Dividing abundance by sampled area does not make any difference for the derived stability and asynchrony metrics. Dividing these metrics by sampling area provides an effective way to standardize by sampling effort. We also refer to the clarifications provided to

Reviewer #2 on this aspect of the analysis. Specifically, we have expanded and clarified the use of an offset and the alternative of analyzing log-response ratios (each response variable divided directly by sampled area) without the offset. In contrast to generalized models, the two approaches are equivalent for linear models (Crawley, MJ. 2013. The R Book Second Edition, p. 415 and p. 566). We show this equivalence by expanding the analysis of log-response ratios to all the variables in the community-level analysis (Supplementary Fig. 5). The results are very similar to those of Fig. 2 in the main text and in Supplementary Fig. 1. We still opt for using the results of models including the offset since these provide a better visualization compared to log-response ratios (compare Fig. 2b-k and Supplementary Fig. 1 with Supplementary Fig. 5). We have articulated these aspects of the analysis in the main text (lines 169-172) and in the Methods (lines 695-706).

I'm still not 100% convinced that the fact these surveys have different target groups of spp does not affect the results. I appreciate the estimation of completeness is a robust way to assess that different survey areas are not affecting the degree to which the richness of the surveyed community is captured but does this not depend completely on the taxonomic scope of the survey in the first place. So you could get similar levels of completeness for surveys which look to assess Scarids versus Acanthurids only. Given these different surveys seem to have, in some cases, quite different target lists, they could all be doing an equally good job of achieving saturation on the rarefaction curve, and thus good completion scores, but that may not mean you can compare metrics which assume the whole assemblage was surveyed. And as I said, the worry here is that overall richness, and by extension functional redundancy, do affect the metrics you derive on stability...at least I believe that was said. So I think it would be important to verify that the differences in overall richness or perhaps just functional richness, of the target list for these surveys is not affecting these results in any way. Of paramount importance would be the OA versus MPA comparisons but perhaps the biggest effects might be on the analysis which look at the geographical spread of metrics and the relationships to MHWs, which may also have geographic bias to them, given different areas tend to be covered exclusively by one survey type or another. It would be good to see some summary of the taxonomic scope of these different surveys and then some analysis to verify these differences are not affecting the results.

We agree with the reviewer that sample completeness does not address the problem of variation in taxonomic scope among monitoring programs. We have acknowledged this facet of our data in the main text (lines 172-174) and in the Methods (lines 781-787) and provided a sensitivity analysis by excluding monitoring programs that targeted a limited set of species (50 or less). We focused on one key result, the positive effect of MPAs on alpha and species stability, asynchrony and functional richness with intensifying MHWs. We found similar results of those in the main analysis (Supplementary Fig. 5), indicating that our findings are not biased by monitoring programs that targeted only a pre-determined subset of the species.

REVIEWERS' COMMENTS

Reviewer #1 (Remarks to the Author):

I have read the authors responses and the revision of this manuscript and I think that, the authors have adequately addressed all my concerns.

Reviewer #2 (Remarks to the Author):

The authors made a lot of effort to address/clarify my concerns. Even though I was not fully convinced that the approach was robust enough to control for sampling effort, I would not insist on my concern on this point given it is common that different thoughts on the same issue. I have no other concerns.

Reviewer #3 (Remarks to the Author):

I thank the authors for conducting the additional sensitivity analysis which excluded surveys with small/specific target species lists. The outcomes of this have assuaged my concerns regarding the potential effect of variable taxonomic scope amongst the different surveys on the key findings here. I have no further comments.